# Towards Conceptualization of "Fair Explanation": Disparate Impacts of anti-Asian Hate Speech Explanations on Content Moderators

**Tin Nguyen,**[†*] **Jiannan Xu,**[‡*] **Aayushi Roy,**[†] **Hal Daumé III,**[†,§] **Marine Carpuat**[†]

[†]Department of Computer Science, University of Maryland
[‡]Robert H. Smith School of Business, University of Maryland
[§]Microsoft Research
{tintn, jiannan, aroy2530, hal3, marine}@umd.edu

## Abstract

Recent research at the intersection of AI explainability and fairness has focused on how explanations can improve human-plus-AI task performance as assessed by fairness measures. We propose to characterize what constitutes an explanation that is itself "fair" – an explanation that does not adversely impact specific populations. We formulate a novel evaluation method of "fair explanations" using not just accuracy and label time, but also psychological impact of explanations on different user groups across many metrics (mental discomfort, stereotype activation, and perceived workload). We apply this method in the context of content moderation of potential hate speech, and its differential impact on Asian vs. non-Asian proxy moderators, across explanation approaches (saliency map and counterfactual explanation). We find that saliency maps generally perform better and show less evidence of disparate impact (group) and individual unfairness than counterfactual explanations.[1]

**Content warning:** This paper contains examples of hate speech and racially discriminatory language. The authors do not support such content. Please consider your risk of discomfort carefully before continuing reading!

## 1 Introduction

Most work at the intersection of the AI explainability and fairness focuses on how explanations can improve Human-AI task performance regarding some fairness criteria. For example, on a recidivism risk assessment task, Dodge et al. (2019) evaluate whether two global and two local explanation methods influence the perceived fairness of the AI model results. In an NLP context, a comprehensive review by Balkir et al. (2022) lays out how explainability techniques have been developed to tackle different sources of biases, including selection, label, model, semantic, and research design biases. This line of research uses explanations as a means to improve fairness measures, but does not consider what constitutes "fair explanations" – namely, explanations that do not bring disparate harm *in and of themselves* to different users or groups of users.

To understand the impact of explanations on different groups, we situate our work in the context of hate speech detection. Here, the human-plus-AI system consists of content moderators who check the potentially hateful content flagged by an AI system, and decide whether to keep or delete it. If the AI predictions came with explanations, this process might be faster and less tiring for moderators, and might alleviate their mental discomfort by allowing them to focus less on hateful content. However, the impact of such explanations might differ across moderators, particularly for those who belong to groups that are targets of the hate speech. Empirical studies are needed to understand these effects, and to ensure that explanation methods used do not disproportionately harm one group over another.

This paper contributes an evaluation of the fairness of explanations by measuring whether they have a disparate impact on different groups of content moderators. We investigate how two types of NLP explanations impact people deciding whether a tweet contains hate speech directed at Asian people. We use saliency maps and counterfactual explanations and measure their impacts using five metrics: classification accuracy, label time, mental discomfort, perceived workload, and stereotype activation (increase in agreement with implicitly biased statements). We hypothesize that people who are targets of the hate speech will experience higher mental costs, and thus measure each metric across races (Asian vs non-Asian).

We find that saliency map explanations simultaneously lead to better task performance and less unfairness than counterfactual explanations.

---

[*]Both authors contributed equally to this work.

[1]The code and human study data are available at https://github.com/jiannan-xu/EMNLP23_Fair_Explanation.

## 2 Background

We review work on explanations in the context of the content moderation and fairness literature.

**Content Moderation.** Researchers have looked for ways to support content moderators in their challenging job by addressing not only their workload, but also their mental discomfort. Exposure to harmful content has been shown to have long-term, negative psychological impact (Newton, 2019; Steiger et al., 2021; Cambridge Consultants, 2019). While entirely preventing exposure is currently impossible (Steiger et al., 2021), it can be minimized by having moderators review initial predictions made automatically (Cambridge Consultants, 2019). Explanations have been used in these settings to explain why a social media post is predicted as harmful, for instance using a dashboard of feature importance (Bunde, 2021), or natural language explanations from ChatGPT (Huang et al., 2023). Such studies evaluate how people perceive these explanations in terms of clarity and usefulness. As a result, we focus on neglected human-centered evaluation dimensions, such as mental discomfort, and verifiable vs. subjective workload.

**Fairness.** Fairness consideration in Explainable AI mostly take the form of measuring the impact of explanations on fairness metrics. For instance, Angerschmid et al. (2022) show that feature-based input influence explanations improve perceived fairness in healthcare. Goyal et al. (2023) show that disclosure of a high correlation between a non-protected feature that is used in the model – e.g. university – and a protected feature – e.g. gender – improves a group fairness metric in a micro-lending task. However, to the best of our knowledge, prior fairness studies have not considered the fairness of the explanations themselves, including whether they have a disproportionate impact on explanation readers across demographic groups.

**Explanation Methods.** Among the wealth of recently proposed explainable NLP methods (Danilevsky et al., 2020), we select two local explanation methods that each represent a distinct family of strategies for surfacing aspects of text that might be indicative of hate speech. Local explanations methods are better suited than global explanations in the content moderation setting where understanding individual (per-tweet) predictions is a priority compared to describing the average behavior of an AI model. Noting such considerations, first, we use saliency map, which highlights the features in the input that are most important for each prediction and is therefore intuitive for laypersons. We generate saliency maps with LIME (Ribeiro et al., 2016), a perturbation-based method which determines input feature salience for any black-box classifier by approximating its behavior locally with a linear model. Second, we use counterfactual explanations, which are minimally modified inputs which flips the prediction. Counterfactual explanations can be viewed as real-life suggested modifications for users to make their tweets no longer hateful. Counterfactual explanations can be generated using language models. Wu et al. (2021) introduce the Polyjuice system to control for different types of counterfactuals and the tokens to be edited. This approach can generate counterfactual explanations for hate speech detection by replacing offensive words or factually unsubstantiated claims.

## 3 Approach: Human Study

We conduct an online human study to quantify the verifiable and subjective impacts of two explanation styles on participants. Simulating the human review of tweets flagged by a hate speech detector, participants know that they are presented with tweets automatically classified as hateful, and they are asked to decide whether each tweet is actually hateful or not. They make this decision without any additional information in the baseline condition, and are presented with either a saliency map (Figure 1) or a counterfactual explanation (Figure 2) in the treatment conditions.

We study two main Research Questions (RQs):

**RQ1:** Does either saliency map or counterfactual explanation influence content moderators on measures related to the psychological impact or efficiency of their task performance?

**RQ2:** When such impact of an explanation style exists, is it "unfair" across groups/individuals?

We quantify the effect of each explanation style via a set of verifiable and subjective metrics. The verifiable metrics are inferred directly based on participants' performance in the main task (hate speech prediction). For the subjective metrics, we measure a set of metrics before and after the main task (when appropriate) to observe how the psychological state of each participant has changed.

This study has been approved by the UMD Institutional Review Board (IRB package: 1941548-3).

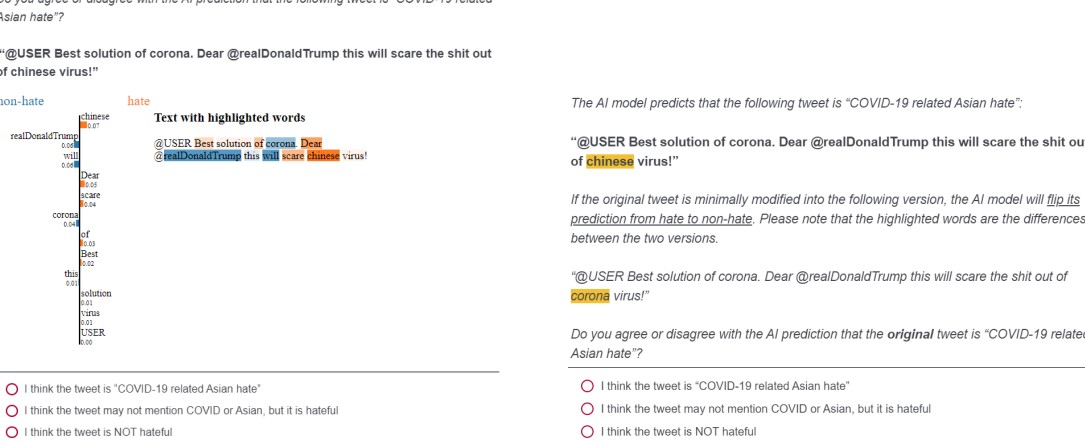

| Figure 1: Saliency Map Interface | Figure 2: Counterfactual Explanation Interface |

## 3.1 Metrics

We develop five metrics: two verifiable (accuracy and label time) vs. three subjective (mental discomfort, stereotype activation, and perceived workload) to evaluate the impact of each explanation style and any differences across racial groups.

**Accuracy.** To get a verifiable measure of hate speech prediction accuracy, we use the ground truth labels by He et al. (2021). We combine their counter-hate and neutral classes into a non-hate class, and provide their definition of "COVID-19 related Asian hate" to our participants: "antagonistic speech that is directed towards an Asian entity (individual person, organization, or country), and others the Asian outgroup through intentional opposition or hostility in the context of COVID-19." We frame the annotation task as a three-way choice between anti-Asian hate, hate (but not anti-Asian), or non-hate, and consider the first two as positive judgements of hate speech.[2]

**Mental Discomfort.** For measuring the mental discomfort (also known as "negative affect" in the psychology literature) one may experience reading potential hate speech, we follow Dang et al. (2018) and Das et al. (2020) and use the Scale of Positive and Negative Experience (SPANE) (Diener et al., 2009). SPANE includes six dimensions for negative affect: Negative, Bad, Unpleasant, Sad, Afraid,

Angry and has been empirically tested on a Chinese population (Li et al., 2013), the main hate speech target here. In our survey, we ask participants to rate their current feeling on a five point scale with respect to each of the six dimensions, both at the start and end of the study. We sum across the six dimensions to get an aggregate negative affect score, as done by Diener et al. (2009), at both time points, and calculate their difference as the mental discomfort metric due to participation in the labeling task.

**Stereotype Activation.** We are interested in whether the hate speech labeling task influences the mental model of participants. Specifically, we ask whether there is an increase in agreement with implicitly biased statements against Asian people after being exposed to content that contains implicit bias against that group. Several studies from psychology literature have studied similar phenomena under the name "stereotype activation" (Moskowitz, 2010; Wheeler and Petty, 2001).

To quantify stereotype activation ($SteAct$), we ask participants at the start to rate on a five point scale how much they (dis-)agree with three implicit anti-Asian statements.[3] At the end of the survey, we have the same question with three paraphrased (semantically identical) but reshuffled statements. Consistent with the activation literature, we assume that exposure to potentially biased content can only *increase* one's mental stereotyping. We therefore calculate changes in agreement to stereotyped questions as the non-negative part of the difference in

---

[2]We initially planned to ask participants to classify tweets as hate or non-hate given AI predictions and explanations. However, based on participant rationales from our pilot study, we realized some participants labeled tweets as non-hate speech only because they did not see an explicit mention of "Asian" or "COVID" in the tweet. We introduced the three-way categorization to address this.

[3]Statements: 1. "China and South Korea have bad eating habits, for example: dogs, cats, and horses."; 2. "The Chinese Communist Party (CCP) uses CCTV to spy on innocent Chinese people."; 3. "The China Communist Party (CCP) and President Xi need to apologize to the world."

agreement levels between the questions at the end of the task ($S'_{1:3}$) and those at the start ($S_{1:3}$) as:

$$SteAct = \frac{1}{3} \sum_{i=1}^{3} \max \left\{ 0, S'_i - S_i \right\} \qquad (1)$$

**Perceived Workload.** To measure a participant's perceived amount of effort, we use the NASA Task Load Index (NASA-TLX), which has six dimensions: mental, physical, and temporal demands, frustration, effort, and performance (Hart, 2006). At the end of the study, a participant rates each dimension on a seven point scale. We take the sum across the six dimensions as perceived workload.

**Label Time.** To obtain a verifiable measure of workload, we calculate the amount of time spent on the twelve hate speech prediction task pages.

## 3.2 Experimental Conditions

We have 3 conditions: baseline (no explanation), saliency map, and counterfactual explanation.

For counterfactual explanations, we find that Polyjuice outputs distort the semantic meaning of the tweets or do not even make sense while counterfactuals generated by ChatGPT are over-corrective and deviate greatly from the original context of the hate-classified tweets as shown in Appendix A.

Noting those constraints, to disentangle the effect of low-quality counterfactual candidates on any observed impact of counterfactual explanations, we manually make our own counterfactual candidates and run them through the fine-tuned RoBERTa classifier to see which of them is the least modified counterfactual that will flip the label from hate to non-hate, which we use as counterfactual explanation. Although the use of human-generated counterfactual candidates limits the generalizability of our findings, our study serves as a useful Wizard-of-OZ experiment to inform future research on which NLP explanation style warrants more work to develop and evaluate "fair explanations". To select which hate-classified samples to generate explanations for and show participants, we selected from the RoBERTa test set the 13 hate-classified samples with the lowest model confidence (i.e. probability) score as they represent a relatively balanced distribution of hate v.s. non-hate ground truth labels. After creating two counterfactuals per tweet, we ran those counterfactuals as additional test data through the RoBERTa classifier again and used the counterfactual that flipped the AI prediction from hate to

non-hate as a counterfactual explanation (in case both counterfactuals of a tweet flipped the label, we selected the first or less modified counterfactual). As an outlier, one tweet had both counterfactuals still classified as hate even though the tweet itself is not hate speech ('Shut up. Taiwan is not China'), so we took it out, leaving 12 tweets, of which 6 are hate and 6 are non-hate according to the ground truth, to show participants.

## 4 Experimental Set-up

### 4.1 Dataset

We use the COVID-HATE dataset of 2290 tweets scraped from Twitter and manually annotated by He et al. (2021) with three labels: 0 (neutral), 1 (counter-hate or Asian-supportive), and 2 (hate speech). The annotation in COVID-HATE was claimed to be performed by two undergraduate students after a rigorous onboarding process and He et al. (2021) only kept the labels that both students agreed on (roughly 2/3 of the original annotations), so upon combining their first two classes into a non-hate class, we use their mapped labels as ground-truths for training and evaluation in our study.

### 4.2 Model configuration

We fine-tune a RoBERTa-based binary classifier for the hate speech classification task. The training details are provided in Appendix B. Since the hate speech label is imbalanced (non-hate: 81.3%, hate: 18.7%), we do a stratified splitting that generates 80% of the original data for training and 20% for testing. The model performance is reasonable as measured by precision (0.7721), recall (0.7093), F-1 score (0.7394) for anti-Asian tweets detection. The model achieves an AUROC score of 0.9273 and an AUPRC score of 0.8399.

### 4.3 Logistics

To avoid the coupling effect of explanation types and minimize the learning effect, we conduct a between-subjects human study via Prolific and develop our survey in Qualtrics.

After a set of initial questions as initial data points for stereotype activation and mental discomfort, we use a Qualtrics Randomizer to randomly assign each participant to one of the three explanation conditions and get relatively equal numbers of samples across conditions. After participants complete the 12 hate speech prediction tasks, they are redirected to a shared-across-conditions set of

ending questions to measure stereotype activation, mental discomfort, and perceived workload.

Using a detailed filtering scheme, we get 283 Prolific participants in our data analysis with relatively equal distributions across explanation conditions and racial groups (Asian v.s. non-Asian). Other details, such as the pay rate, participants filtering criteria, baseline condition interface, and the questions supporting the subjective metrics, can be found in Appendix C and Appendix D.

## 5 Findings

After computing the five metrics of interest per participant, we analyze the distribution of responses—across explanation style and participant race—, visualizing them using box and whisker plots, and comparing the means using t-tests. We only report findings that achieve statistical significance from the t-tests (p-value $\leq 0.05$).[4] Since p-value is an indicator of whether an effect exists but not of its magnitude (Wasserstein and Lazar, 2016), we also report Cohen's d score (Cohen, 2013) and its 95% confidence interval from bootstrapping.

Our results indicate that statistically significant effects presented in the paper have sensible Cohen's d score estimates (small to moderate effects that require statistical methods to be observed (Cohen, 2013)) and reasonable 95% confidence intervals. Statistically insignificant or less interesting results for the remaining metrics are shown in Appendix G. To examine varying treatment effects for individuals or subgroups in a population, we analyze the heterogeneous treatment effects in Appendix F.

### 5.1 Saliency map outperforms counterfactual explanation on three overall metrics

We first look at how each explanation style performs across our metrics in general and find two significant results about the utility of saliency map.

**Mental Discomfort.** As shown in Figure 3, saliency map (M = 1.7010, SD = 5.0076) yields significantly lower mental discomfort than the no explanation baseline (M = 3.4194, SD = 5.9121), t(188) = -2.1535, p-value = 0.033, Cohen's d = -0.3142, 95% CI = [-0.5915, -0.0362]. A plausible mechanism which may explain this observation is that saliency map gives participants the option to just look at the top few most "hateful" tokens without necessarily putting them into context of a whole

sentence, therefore reducing the need to comprehend the discriminatory intent of the tweet and thus lowers the mental discomfort they experience.

**Accuracy.** As shown in Figure 4, saliency map (M = 0.4759, SD = 0.1036) yields significantly higher hate speech prediction accuracy than counterfactual explanation (M = 0.4651, SD = 0.1088), t(188) = 2.3078, p-value = 0.022, Cohen's d = 0.3367, 95% CI = [0.0518, 0.6328]. This result is consistent with the psychology literature: Reinhard et al. (2011) find that people make more accurate lie predictions (whether someone is lying) in settings where they have higher "situational familiarity". Although lie and hate speech are not the same, we see several shared attributes of these two concepts, e.g. often originating from bad intent, containing wrong facts. Furthermore, they find that the gain in accuracy is thanks to verbal content cues rather than non-verbal content cues. This verbal setting is similar to our online context. Therefore, we expect their findings to be relatively generalizable to our study, i.e. if we can show that people have higher "situational familiarity" to saliency map than to counterfactual explanations, Reinhard et al. (2011) will help explain why crowdworkers gain higher accuracy when given saliency map. The missing piece of our argument can be found in Yacoby et al. (2022), who show that judges are at first confused by and later ignore counterfactual explanations in AI-assisted public safety assessment (PSA) tasks. Instead, they prefer looking at each defendant's specific features, which analogously corresponds to saliency map in our study. Altogether, Yacoby et al. (2022) show that humans have higher "situational familiarity" with saliency map than counterfactual explanations, from which Reinhard et al. (2011) show that higher "situational familiarity" leads to higher classification accuracy of verbal content.

**Stereotype Activation.** As shown in Figure 5, counterfactual explanation (M = 0.1900, SD = 0.3365) yields significantly higher stereotype activation than the baseline (M = 0.0968, SD = 0.1990), t(184) = 2.2861, p-value = 0.023, Cohen's d = 0.3371, 95% CI = [0.0648, 0.5884]. One reason is that although a counterfactual may make the AI model predict *non-hateful*, the counterfactual itself is not necessarily non-hateful. For example, the counterfactual explanation for the tweet "@USER Pussies.. That's what the Chinese are known for... retreat Losers!!! #ChineseVirus" is

---

[4]We report results of t-tests in APA format.

still arguably hateful even after removing the "#ChineseVirus" hashtag. Therefore, the false prediction by the AI model that a still-hateful counterfactual is non-hateful may bias participants into thinking that the underlying discriminatory content of such counterfactuals are acceptable, thereby increasing their stereotype activation. We might view counterfactual explanations as an analogy to "counter-stereotypic images" from social psychology, where Nelson and Kinder (1996) find that participants who are shown counterstereotypic images are influenced by racial attitude more than a control group who is shown nothing. Analogously, counterfactual explanations might trigger people's implicit bias.

To understand why counterfactual explanations seem less desirable overall than salience maps, we recall a study of Laugel et al. (2019), who find that post-hoc counterfactual explanations, which "use instances that were not used to train the model to build their explanations", will risk not satisfying desirable properties of AI explanations such as "proximity, connectedness and stability". Since the counterfactual explanation in our study is post-hoc, that study might also be applicable to our context.

## 5.2 Counterfactual explanation yields disparate impact on two metrics, but saliency map is not completely "fair"

We break down the explanation impact results by whether a crowdworker identifies as Asian or not to investigate potential disparate impacts of explanation styles across impacted populations.

**Mental Discomfort.** As shown in Figure 6, we observe significant disparate mental discomfort: Asian participants (M = 4.1087, SD = 5.2138) experience significantly higher mental discomfort than non-Asian participants (M = 1.7447, SD = 5.8273) in the counterfactual explanation condition, t(91) = 2.0370, p-value = 0.045, Cohen's d = 0.4271, 95% CI = [0.0465, 0.8022], but not in other conditions. Our potential justification comes from Hegarty et al. (2004) who find that when asked to provide counterfactual explanations on how to make a hypothetical male (either gay or straight) feel less discomfort in a bar dominated by males from the opposite sexual orientation, humans are strongly influenced by hetero-centric norms. If this finding generalizes from sexual orientation to race, given counterfactual explanations, all participants may experience white-centric norms, but

those norms pressure Asians more than non-Asians (the white-majority group), justifying the disparate mental discomfort.

**Label Time.** As shown in Figure 7, counterfactual explanations also give significantly disparate impact in terms of verifiable workload (measured through label time), p-value = 0.012, Cohen's d = 0.5365, 95% CI = [0.1451, 0.9367]. However, the disparate impact here is in the opposite direction: disadvantaging non-Asian participants (M = 349.3864, SD = 196.7625) rather than Asian participants (M = 256.9960, SD = 142.8468). Many Asian participants or people in their circles may have first-hand experience with anti-Asian hate speech, making them more efficient in formulating their own predictions about such content. If we come back to the "situational familiarity" concept from Reinhard et al. (2011) which we have argued in section 5.1 to be low for counterfactual explanation, we can argue that because participants get almost no "situational familiarity" from the counterfactual explanations, they can only rely on their "situational familiarity" with the anti-Asian hate speech prediction task itself (absent any explanations), which many Asians have practiced during the pandemic to identify threats against themselves. As Asians have higher "situational familiarity", they may perform the prediction task faster than non-Asians, who might still be confused due to low "situational familiarity" with the counterfactual explanations and the anti-Asian hate speech prediction task itself.

Social media platforms face an ethical vs. economic tradeoff when allocating hate speech prediction tasks to their content moderators. Suppose a platform is expected to provide counterfactual explanations to its content moderators, it might be incentivized to give those hate-classified tweets to content moderators who belong to the targeted minority to minimize label time and thus the annotation cost pet tweet, at the expense of disproportionately high mental discomfort experienced by the minority group. Therefore, our finding might inform future AI-assisted decision making regulations, suggesting the introduction of externalities, e.g. fines, to limit (if not completely prevent) social media platforms from allocating AI-classified hateful content and/or "unfair" AI explanations that cause disparate mental discomfort to the targeted minority in their content moderator pool.

Tangentially, our finding might also shed light on the mixed results regarding a data quality dilemma

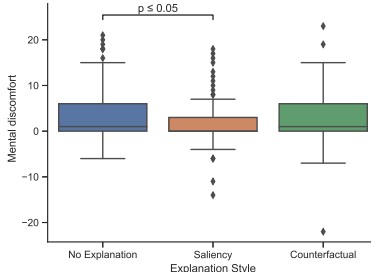 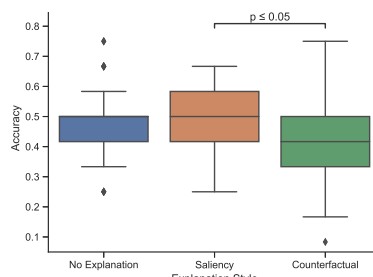 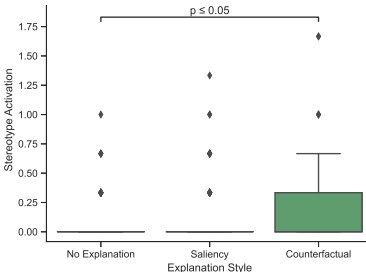

Figure 3: The saliency map condition yields significantly less mental discomfort than the no explanation baseline (p-value = 0.033).

Figure 4: The saliency map condition yields significantly higher hate speech prediction accuracy than the counterfactual explanation condition (p-value = 0.022).

Figure 5: The counterfactual explanation condition yields significantly higher stereotype activation than the no explanation condition (p-value = 0.023).

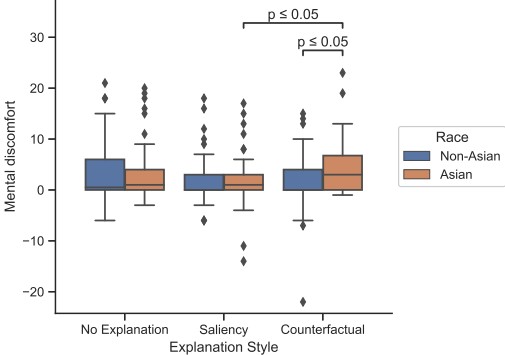 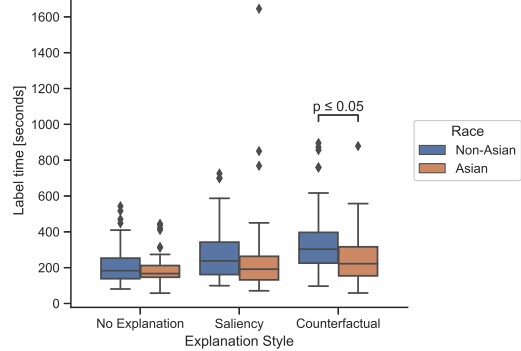

Figure 6: Counterfactual explanation yields significantly disparate mental discomfort against Asians (p-value = 0.045). Asians with counterfactual explanations get significantly more mental discomfort than Asians with saliency map (p-value = 0.035).

Figure 7: The counterfactual explanation condition yields significantly disparate label time against non-Asians. (p-value = 0.012), indicating that sharing the same sensitive feature value as the target minority might make content moderators more efficient.

in the recent hate speech prediction literature. On the one hand, Olteanu et al. (2017) find that people who have experienced online harassment make less accurate hate speech prediction than people who have not. Their finding might suggest an incentive to exclude the targeted minority from the hate speech prediction task. On the other hand, Sap et al. (2019) find that providing crowdworkers with information about the likely race of a tweet's author (based on the dialect of the tweet) will improve data quality of the annotation as crowdworkers will be less likely to annotate tweets with the African American English (AAE) dialect as offensive. If we assume that content moderators from the targeted minority will know best whether the race of the tweets' authors is the same as the targeted minority, their finding might imply an incentive to intentionally include the targeted minority into the hate speech prediction task. Since our study suggests an economic (yet not ethical) incentive for

industry to include the targeted minority in the hate speech prediction task, our finding aligns with Sap et al. (2019), but not Olteanu et al. (2017).

**Stereotype Activation.** Figure 8 shows that saliency map yields marginally disparate stereotype activation against non-Asian participants (M = 0.1736, SD = 0.2965) but not for Asian participants (M = 0.0816, SD = 0.2079), t(95) = 1.7535, p-value = 0.083, Cohen's d = 0.3598, 95% CI = [-0.0419, 0.7586]. This result should not be interpreted as discrediting saliency map, but it is a caveat against thoughtlessly using saliency map without carefully examining its potential disparate impact on explanation readers in a downstream application context.

### 5.3 When explanation is required, which explanation style is better might depend on the pool of explanation readers

With the rapid adoption of AI into decision-making tasks, some jurisdictions have started to require

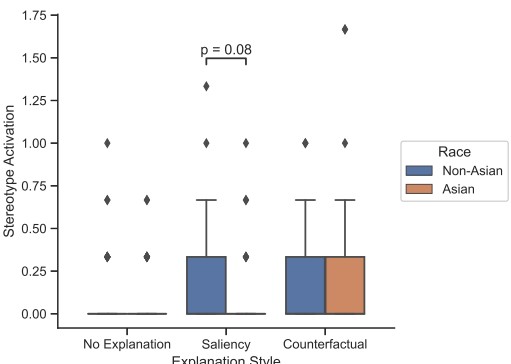

Figure 8: There is evidence with marginal significance that the saliency map condition may yield disparate stereotype activation against non-Asians. (p-value = 0.083).

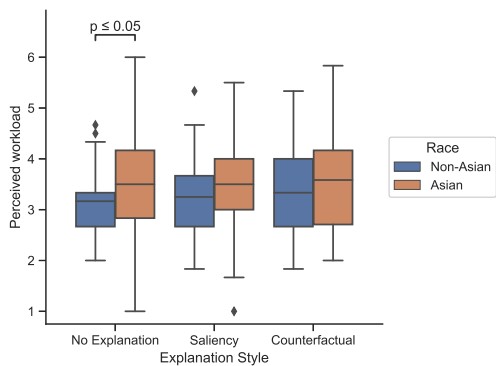

Figure 9: There is significant disparate perceived workload in the baseline condition (p-value = 0.041) but not in either of the saliency map or counterfactual explanation conditions.

AI explanations for humans if the AI is to advise humans in making certain decisions.[5][6]

If AI explanation becomes mandatory one day for content moderation, based on the results from Figure 9, the disparate perceived workload from the no explanation condition is no longer an evident concern. However, a natural question arises: which explanation style to use? Observations in Figure 6 show that if the social media platform has access to content moderators outside the targeted minority, they may have more flexibility in selecting the explanation method because there is no noticeable difference in non-Asian participants' mental discomfort between the saliency map and counterfactual conditions. However, if their pool of content moderators is limited to only the targeted minority (e.g. Asian), they should avoid counterfactual explanation because for the targeted minority, counterfactual explanation yields significantly higher mental discomfort than saliency map.

### 5.4 Qualitative Analysis: Asians who experienced disparate mental discomfort are less likely to leave a rationale

We analyze if the optional free-text rationales give additional insights into the quantitative results above. When asked to explain their ratings for men-

tal discomfort at the end of the survey, participants tended to discuss whether they felt affected by the content of the tweets. For example, P157 who is Asian and saw the saliency map wrote, "Reading some of that stuff left me with some negative emotions", whereas other participants such as P219 who is White and saw the saliency map explanations noted they did not feel affected by the content: "I'm not feeling any of the emotions at this time". Out of the 27 participants who indicated discomfort from the tweets in their rationales, 14 (52%) were Asian. However, only 18% of Asian participants left a negative rationale after experiencing mental discomfort while 24% of white participants left a negative rationale after the experiment. This is despite more Asian participants (21%) leaving any rationale (negative or otherwise) for their mental discomfort rating versus 18% of white participants. Asians are not less likely to leave a rationale altogether, but are less likely to leave one when they have experienced mental discomfort.

Across all conditions, the majority of participants left at least one rationale to explain their decision in tweet labeling, with approximately 20% of participants leaving a rationale for every decision. Participants tended to reference a key word in the tweet (often a racist word) or use their understanding of the provided definition of COVID-19 related hate speech (such as the lack of mentioning Asian people) to explain their decision. Around 45% of the participants did not leave any rationales. As the experiment progressed, participants across conditions became less likely to leave a rationale.[7]

---

[5]Article 13.1 of the EU Artificial Intelligence Act (AIA) requires that: "High-risk AI systems shall be designed and developed in such a way to ensure that their operation is sufficiently transparent to enable users to interpret the system's output and use it appropriately".

[6]The 'Notice and Explanation' section of the US Blueprint for an AI Bill of Rights requires that "Automated systems should provide explanations that are technically valid, meaningful and useful to you and to any operators or others who need to understand the system".

---

[7]These insights remain qualitative only. Performing t-tests and chi-squared tests of independence on the number of ratio-

## 5.5 Individual Fairness: Counterfactual Explanation is more individually unfair

In our preceding experiments, we consider fairness from the perspective of group fairness - is one group (Asian) harmed more than another (non-Asian). The other standard way of measuring fairness is individual fairness: are similarly situated individuals treated similarly. In our context, the question is whether two similar annotators are faced with the same amount of mental discomfort or perceived workload. A key challenge in individual fairness is defining "similar" individuals; we define this as the pairwise distance averaged across responses to 9 task-relevant multiple-choice questions. We can then measure the degree of individual fairness, which captures a notion quite different from group fairness. (See Appendix E for details.)

We find that for Mental Discomfort, introducing either explanation style decreases individual unfairness, which contrasts the individual fairness results for other output metrics. Among the two styles, counterfactual explanation is more unfair. For Stereotype Activation, introducing either explanation style increases individual unfairness. Among the two styles, counterfactual explanation is also more unfair regarding this second output metric. For Perceived Workload, introducing either explanation style increases individual unfairness. With respect to Label time, introducing either explanation style increases individual unfairness.

Comparing our group fairness results with our individual fairness results, we see that counterfactual explanation is more individually unfair than saliency map, which aligns with our previous finding that counterfactual explanation shows more evidence of group-wise unfairness, motivating the use of saliency map if fairness is a priority concern when choosing an explanation style. Tangentially, introducing either explanation style increases individual unfairness with respect to many metrics (stereotype activation, perceived workload, label time) but decreases individual unfairness with respect to mental discomfort. This result questions whether social media should use AI explanation for the hate speech classification task in the first place.

## 6 Conclusion

We conduct a between-subjects human study across three explanation conditions to study potential (dis-

___

nales left by individuals in different conditions or of different races did not find any statistically significant differences.

parate) impacts of saliency map and counterfactual explanations on content moderators, using crowdworkers as a proxy. We find that first, saliency map is the most desirable condition overall. Second, counterfactual explanation exhibits a tradeoff in terms of disparate mental discomfort and disparate label time, thereby highlighting the need for legal intervention to minimize risks of labor abuse against content moderators in the targeted minority. However, saliency map is not necessarily innocent as there is marginal evidence of its disparate stereotype activation. Third, mandatory introduction of explanation may mitigate disparate perceived workload, but careful selection of explanation style may be needed depending on the racial distribution of the content moderator pool. Fourth, Asians who experienced mental discomfort are more reluctant to leave a rationale. Fifth, counterfactual explanation is more individually unfair than saliency map. Our results suggest that even though counterfactual explanations seem less desirable and less "fair" for proxy content moderators, further research is needed to confirm the utility and fairness of saliency map before more adoption of this explanation style by social media platforms.

If follow-up research can validate that our findings are generalizable beyond crowdworkers to real content moderators, new AI-related legislation may incentivize social media platforms to implement an effective and "fair" explanation method, e.g. saliency map, on a larger scale. However, if due to logistic constraints, an overall less desirable or "unfair" style, e.g. counterfactual explanations, must be used, stakeholders might consider a race-conscious content moderation policy, e.g. by assigning content that AI classified as targeting racial minorities to moderators of other races to minimize mental impact. In summary, this work aims to responsibly implement explanations methods in the real world, thereby reducing workload and improving psychological well-being without sacrificing job performance, particularly of those sharing demographic characteristics with hate speech victims.

## Limitations

Our study focuses on how potential content moderators—who would likely be the ones faced with explanations—are impacted by seeing those explanations. However, our participant pool is crowdworkers, standing in as proxies for content moderators, which introduces an ecological valid-

ity concern. As discussed, in the case that lay users (e.g., of social media platforms) are shown explanations, this concern is mitigated because crowdworkers are probably a better surrogate for lay users. Our findings are also limited to one particular demographic—people who self-identify as "Asian" (versus not)—and one particular task ("COVID-19-related Asian hate") drawn from one particular dataset He et al. (2021) in one particular time period (early 2020s) in one particular language (English) and with two particular explanation conditions (plus no explanation). Future work may test the generalizability of our findings in the hate speech prediction context with respect to another demographic feature, e.g. (gender-based) misogyny or transphobia, or (sexual orientation-based) anti-queer hate speech, or other AI-assisted decision making tasks where both saliency map and counterfactual explanations are relevant and where a certain minority that has been historically disadvantaged may also serve as the decision-maker.

Our findings are also only as good as our measurements and the wording of the task and survey questions. We know from previous literature that these can have non-trivial effects on the results. For instance, Giffin et al. (2017) found that simply *naming* an explanation technique with a science-y sounding name increased people's satisfaction with it. During the tutorial of our experiment we named saliency map but not counterfactuals, which may have had a small impact on results.

We have only experimented with two explanations styles (saliency and counterfactual), which may limit the potential for adopting our findings into real-life content moderation. Future work may expand the scope of findings with more explanation styles. For instance, we can test a hypothesis that data influence explanations (Han et al., 2020) might have a more disparate impact against Asian participants in terms of mental discomfort. One potential justification for this hypothesis is that participants will need to read more hateful-classified samples from the training set as explanations, and Asians might experience higher per-instance (and thus significantly higher cumulative) mental discomfort than non-Asians. Another option is global explanations, such as anchors (Ribeiro et al., 2018), which identify tokens likely to cause hate predictions in the entire dataset. It may reduce the amount of time exposed to hate tweets, thereby potentially mitigating disparate workload or mental discomfort. In summary, future work may add more experimental conditions with other explanation styles, e.g. data influence or (global) anchors, if the budget allows.

## Ethical Considerations

Since our study involves tweet samples with derogatory, offensive, and discriminatory languages. We *expect* these to have negative mental impacts on participants, at least in the short term (and we find that they do). To mitigate this harm, we have: (a) given advanced notice to potential participants in Prolific (where they are recruited); (b) repeated that notice in Qualtrics (where the study takes place); and (c) give them the option to terminate the study at at time. As mentioned in the paper, the study was IRB approved.

In the dataset attached to our paper, we delete all personally identifiable information from our human study participants to protect their privacy. For example, we replace actual Prolific IDs of our participants (in the column PROLIFIC_PID, which are used in the individual fairness evaluation notebook) by dummy indices from 1 to 287.

## Acknowledgement

This work was supported in part by U.S. Army Grant No. W911NF2120076. We thank EMNLP Reviewers, Area Chairs and Program Chairs for evaluating our paper with their valuable time! We also thank all members of the UMD CLIP and HCIL labs, especially those who provided feedback to our paper, e.g. Dr. Joel Chan, Do Won Kim, Kyle Seelman, Min Cheong Kim, Naimul Hoque, Navita Goyal, Sandra Sandoval and Sathvik Nair!

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

## A  Details of Counterfactual Explanations

For counterfactual explanations, at the first planned step of generating counterfactuals from Polyjuice, we realized that many of those counterfactuals completely distort the semantic meaning of the tweets or do not even make sense.[8] Another option is to use pre-trained large language models (such as ChatGPT) to generate counterfactuals, but these models have strict content policies against quoting hateful languages in the prompts and even when the models accept our prompts, the counterfactuals they generate are over-corrective and deviate greatly from the original context of the hate-classified tweets.[9]

## B  Details of the AI Hate Speech Classifier

We fine-tune a RoBERTa-based binary classifier for 10 training epochs, on a Quadro RTX 5000 GPU machine, maximum sequence length of 128, batch size of 16, warm-up steps of 500, learning rate of 1e-05, and weight decay of 0.01. We use Roberta-base model, which includes 12 layers, 768 hidden nodes, 12 head nodes, 125M parameters, and add a linear layer with two nodes for binary classification. Training the classifier takes several minutes as the data set is relatively small.

## C  Detailed Human Study Interfaces

We use one definition of "COVID-19 related Asian hate" throughout our study: "antagonistic speech that is directed towards an Asian entity (individual person, organization, or country), and others the Asian outgroup through intentional opposition or hostility in the context of COVID-19."

*Do you agree or disagree with the AI prediction that the following tweet is "COVID-19 related Asian hate"?*

**"@USER Best solution of corona. Dear @realDonaldTrump this will scare the shit out of chinese virus!"**

○ I think the tweet is "COVID-19 related Asian hate"

○ I think the tweet may not mention COVID or Asian, but it is hateful

○ I think the tweet is NOT hateful

Figure 10: Human study interface with an example question (No Explanation condition)

---

[8]For example, Polyjuice generates counterfactuals for the two tweets 'gave my d*ck the coronavirus' and 'We all will call it s Chinese virus' to be 'gave my son the coronavirus' and 'We all will call it s a real christmas virus'.

[9]An example GPT-generated counterfactual to the tweet "@USER Best solution of corona. Dear @realDonaldTrump this will scare the shit out of chinese virus!" is "If Trump had taken a more proactive and coordinated approach to the pandemic from the outset, perhaps the impact of the virus would not have been as severe as it has been."

*Do you agree or disagree with the AI prediction that the following tweet is "COVID-19 related Asian hate"?*

**"@USER Best solution of corona. Dear @realDonaldTrump this will scare the shit out of chinese virus!"**

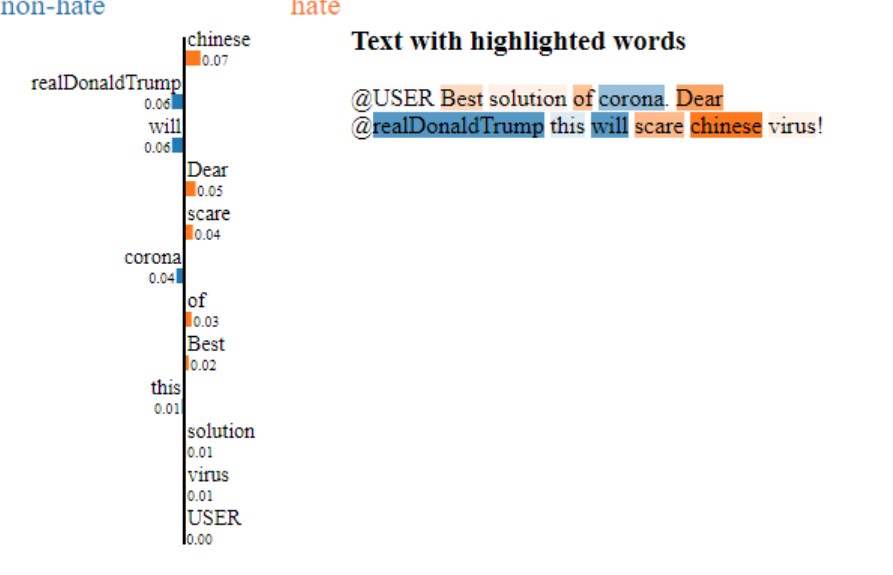

○ I think the tweet is "COVID-19 related Asian hate"

○ I think the tweet may not mention COVID or Asian, but it is hateful

○ I think the tweet is NOT hateful

Figure 11: Human study interface with an example question (Saliency Map condition)

*The AI model predicts that the following tweet is "COVID-19 related Asian hate":*

**"@USER Best solution of corona. Dear @realDonaldTrump this will scare the shit out of chinese virus!"**

*If the original tweet is minimally modified into the following version, the AI model will flip its prediction from hate to non-hate. Please note that the highlighted words are the differences between the two versions.*

*"@USER Best solution of corona. Dear @realDonaldTrump this will scare the shit out of corona virus!"*

*Do you agree or disagree with the AI prediction that the **original** tweet is "COVID-19 related Asian hate"?*

- ○ I think the tweet is "COVID-19 related Asian hate"
- ○ I think the tweet may not mention COVID or Asian, but it is hateful
- ○ I think the tweet is NOT hateful

Figure 12: Human study interface with an example question (Counterfactual Explanation condition)

On a scale from "Definitely not feeling so" to "Definitely feeling so", please rate whether you are feeling each of the following emotions now:

| | Definitely not feeling so | Probably not feeling so | Not sure | Probably feeling so | Definitely feeling so |
|---|---|---|---|---|---|
| Negative | ○ | ○ | ○ | ○ | ○ |
| Bad | ○ | ○ | ○ | ○ | ○ |
| Unpleasant | ○ | ○ | ○ | ○ | ○ |
| Sad | ○ | ○ | ○ | ○ | ○ |
| Afraid | ○ | ○ | ○ | ○ | ○ |
| Angry | ○ | ○ | ○ | ○ | ○ |

Figure 13: Questions to quantify mental discomfort: Scale of Positive and Negative Experience (SPANE)

On a scale from "Strongly Disagree" to 'Strongly Agree", please rate how much you agree/disagree with the following statements:

| | Strongly Disagree | Disagree | Neutral | Agree | Strongly Agree |
|---|---|---|---|---|---|
| The world needs an apology from President Xi and the China Communist Party (CCP). | O | O | O | O | O |
| Dogs, cats, and horses are victims of bad eating habits in China and South Korea. | O | O | O | O | O |
| This is an attention check - select "Strongly Agree" for this statement. | O | O | O | O | O |
| Innocent Chinese people are spied on by the Chinese Communist Party (CCP) with CCTV. | O | O | O | O | O |

Figure 14: Questions to quantify stereotype activation: rating on a five-point scale how much participants (dis-)agree with three implicit anti-Asian statements

On a scale from 1 to 7, where 1 is "Not at all" and 7 is "Completely", please rate your answer to each of the following questions:

| | 1 - Not at all | 2 | 3 | 4 | 5 | 6 | 7 - Completely |
|---|---|---|---|---|---|---|---|
| Mental Demand: How mentally demanding was the task? | O | O | O | O | O | O | O |
| Physical Demand: How physically demanding was the task? | O | O | O | O | O | O | O |
| Temporal Demand: How hurried or rushed was the pace of the task? | O | O | O | O | O | O | O |
| Performance: How successful were you in accomplishing what you were asked to do? | O | O | O | O | O | O | O |
| Effort: How hard did you have to work to accomplish your level of performance? | O | O | O | O | O | O | O |
| Frustration: How insecure, discouraged, irritated, stressed, and/or annoyed were you? | O | O | O | O | O | O | O |

Figure 15: Questions to quantify perceived workload: NASA Task Load Index (NASA-TLX)

## D   Human Study Pay Rate and Participants Selection Criteria

We set our Prolific pay rate at $12 per hour. We estimate the total study duration based on the completion duration statistics of the most time-consuming condition from our pilot experiment, and set the estimated duration for our main study across conditions to be 14 minutes, corresponding to a pay amount of $2.80 for every condition.

In Prolific, we apply the following filters for participants: Living in the USA, English fluency, Prolific approval rate $\geq 98\%$. To get roughly the same number of Asian and non-Asian participants, we set up two almost identical Prolific studies with only one difference in the ethnicity filter: one includes only 'Asian' and the other includes 'Black', 'White', and 'Other' (i.e. non-Asian).[10]

We get 287 Prolific participants in total. To ensure data quality and consistency with Prolific policy, we exclude the 4 participants who fail at least two attention checks from our data analysis, leaving us with 283 data points.

## E   Detailed Individual Fairness Evaluation

**Concept**   There are two major popularized schools of fairness in the AI literature: group fairness and individual fairness. The main idea of group fairness is that outcomes (values with respect to an output feature of interest, e.g. accuracy) should be relatively equalized, or should not differ significantly across different (demographic) groups. Otherwise, there is disparate impact (Barocas et al., 2017).

Another popular perspective in the fairness literature is "individual fairness", with the intuition that the outcomes are fair if similar individuals get similar outcomes (Dwork et al., 2012). If we can apply this intuition of "individual fairness" into a quantifiable metric, we might evaluate which of the three explanation conditions are more individually unfair with respect to each of the five output metrics (e.g. mental discomfort). Our motivation is that if the new "individual fairness" findings also aligns with the previous "group fairness" findings, e.g. if "counterfactual explanations" are more unfair than "saliency map" from both group fairness and individual fairness perspectives, the new result will further enhance the Soundness of our claims.

**Evaluation Pipeline**   One major challenge in applying the "individual fairness" intuition is to define a task-relevant pairwise distance function to calculate how "similar" any two individuals are. To address this issue, at the start of the survey, we ask each participant 9 multiple choice questions about their individual input features related to the anti-Asian hate speech classification task. We design the answer choices to each question such that nearer-located answers choices (e.g. A and B) are semantically closer than farther-located answer choices (e.g. A and C). The list of multiple-choice questions are below.

**Question 1**: Have you ever been a victim of online hate speech (if multiple options apply, please choose the first applicable option)?

A. Yes: online hate speech against Chinese

B. Yes: online hate speech against Asian but not Chinese

C. Yes: online hate speech against a non-Asian race/ethnicity (such as against Black, Hispanic, etc.)

D. Yes: online hate speech based on a non-race sensitive attribute (such as gender, sexual orientation, etc.)

E. No

F. Prefer not to answer

**Question 2**: Have you ever been a victim of verbal (in-person) hate speech (if multiple options apply, please choose the first applicable option)?

A. Yes: verbal hate speech against Chinese

B. Yes: verbal hate speech against Asian but not Chinese

C. Yes: verbal hate speech against a non-Asian race/ethnicity (such as against Black, Hispanic, etc.)

---

[10]Since a participant cannot select more than one racial option in their Prolific registration and we exclude the 'Mixed' race category, assuming all participants' registrations were authentic, there should be no misassignment or leakage of participants between the Asian and non-Asian versions.

D. Yes: verbal hate speech based on a non-race sensitive attribute (such as gender, sexual orientation, etc.)

E. No

F. Prefer not to answer

**Question 3**: How much time in total have you spent visiting and/or living in Asia?

A. Never

B. 1 day - 1 month

C. 1 month - 1 year

D. 1 year - 5 years

E. Over 5 years

F. Prefer not to answer

**Question 4**: Were you born in the USA, and if not, at what age did you move to the USA?

A. Born in the USA

B. Moved to the USA at an age lower than 5 years old

C. Moved to the USA at an age between 5 and 18 years old

D. Moved to the USA at an age between 18 and 30 years old

E. Moved to the USA at an age higher than 30 years old

F. Prefer not to answer

**Question 5**: Please choose the option that best describes your family background (if multiple options apply, please choose the first applicable option):

A. Mainland China

B. Taiwan, Hong Kong, or Macau

C. A Sinosphere country (Japan, North/South Korea, or Vietnam)

D. An East Asian or Southeast Asian country not mentioned above (such as Mongolia, Singapore, etc.)

E. An Asian country outside East/Southeast Asia

F. No Asian background

G. Prefer not to answer

**Question 6**: How many Asian languages (such as Mandarin Chinese, Hindi, Vietnamese, etc.) do you speak?

A. 0

B. 1

C. 2

D. 3 or more

E. Prefer not to answer

**Question 7**: How necessary do you think online content moderation is (when it is weighed against other rights such as free speech)?

A. Very unnecessary

B. Unnecessary

C. Neutral

D. Necessary

E. Very necessary

**Question 8**: What do you personally think should be the highest appropriate sanction against the most extreme online hate speech creators?

A. No sanction

B. Deleting the hateful content (such as specific hate tweets)

C. Banning the hate speech creator from the relevant social media platform

D. Civil damages (such as financial compensation for the victims' mental sufferings)

E. Criminal punishment (such as community service, probation, jail time)

**Question 9**: How often do you think that you "have no place in this world", "feel left out", or "feel like an outsider"?

A. (Almost) always

B. Daily

C. Weekly

D. Monthly

E. Yearly

F. (Almost) never

We filter out any "Prefer not to answer" answers and map the remaining multiple-choice answers for each question to normalized, equidistant values between 0 and 1 (e.g. A to 0, B to 0.25, C to 0.5, D to 0.75, E to 1). Next, we compute the pairwise distance (absolute difference, averaged across the 9 individual input features above) for every pair of individuals within a given explanation condition. Finally, we develop a simple and interpretable pipeline to compute an "individual unfairness" metric, as follows:

**Pipeline: Individual UnFairness Evaluation**

Given output feature X (e.g. mental discomfort) and explanation condition i (e.g. saliency map):

**Step 1**: Find the top k pairs of most similar two individuals (with smallest pairwise average distances)

**Step 2**: Within the top k pairs, for the two individuals A and B in each pair with $avg\_distance(A, B)$, i.e. distance averaged across their 9 individual input features, to get a $pairwise\_individual\_unfairness$ metric for A and B, we compute the absolute difference in their output values weighted by $[(1 - avg\_distance(A, B)]$. Our rationale is that the more similar two individuals are (i.e. the smaller their $avg\_distance$ is), the more "individually unfair" it will be for their output values to differ.

$$pairwise\_individual\_unfairness(A, B) = [1 - avg\_distance(A, B)] \cdot |output(A) - output(B)|$$

**Step 3**: Calculate the mean of pairwise individual unfairness scores across the k pairs to use as the individual unfairness score of explanation condition i with respect to output feature X.

**Step 4**: Perform t-tests between the distributions of output feature X in explanation condition i and the same output feature X in another explanation condition (such as j) to compare the "individual unfairness" between conditions i and j with respect to the feature X.

**Detailed Results**    The results of the individual (un)fairness evaluation pipeline above are summarized in Table 1. In particular, to ensure that our findings are stable, we vary the number of top pairs considered ($k$) as a hyperparameter with values ranging from {100, 200, 400, 1000, 2000, 4000}. We report the "individual unfairness" cross-conditions comparisons which remain statistically significant (p-value < 0.05) across multiple settings of k.

| Metric for Individual Unfairness evaluation | Comparison (p-value < 0.05) of Individual Unfairness (IU) | $k$ (number of evaluation pairs giving significant Comparison) |
|---|---|---|
| Mental Discomfort (MD) | MD_IU(SM) < MD_IU(NE) | 400, 1000, 2000, 4000 |
| | **MD_IU(CE) > MD_IU(SM)** | 400, 1000, 2000, 4000 |
| | MD_IU(CE) < MD_IU(NE) | 1000, 2000, 4000 |
| Stereotype Activation (SA) | SA_IU(SM) > SA_IU(NE) | 1000, 2000, 4000 |
| | **SA_IU(CE) > SA_IU(SM)** | 2000, 4000 |
| | SA_IU(CE) > SA_IU(NE) | 1000, 2000, 4000 |
| Perceived Workload (PW) | PW_IU(SM) > PW_IU(NE) | 100, 200, 400, 1000, 2000, 4000 |
| | PW_IU(CE) > PW_IU(NE) | 100, 200, 400, 1000, 2000, 4000 |
| Label Time (LT) | LT_IU(SM) > LT_IU(NE) | 100, 200, 400, 1000, 2000, 4000 |
| | LT_IU(CE) > LT_IU(NE) | 100, 200, 400, 1000, 2000, 4000 |

Table 1: Individual UnFairness comparison per output metric across conditions (NE: No Explanation; SM: Saliency Map; CE: Counterfactual Explanation). **Bold** comparisons are to check which explanation style (SM or CE) is more individually unfair.

## F  Heterogeneous Treatment Effects: Counterfactual Explanations are group-wise unfair with respect to race, gender, and age

We have explored the treatment effects separately for different groups, testing if the difference in treatment effects is statistically significant. However, there are potential problems with this approach. Specifically, the traditional sample split cannot capture higher-order interactions between treatment and baseline characteristics. For example, the most and least affected individuals might not fall on the two ends of one characteristic (Asian v.s. non-Asian). Instead, the most affected group may be a segment of the population that is defined by a group of characteristics in a non-linear way.

To better understand how did the treatment effect vary across individuals and their characteristics, we adopt a machine learning approach to investigate the heterogenous treatment effects. We estimate heterogeneous treatment effects using Double Machine Learning (Chernozhukov et al., 2018). We use *Tree Interpreter* to provide a presentation-ready summary of the key features that explain the biggest differences in responsiveness to an intervention.

Specifically, we find that regarding stereotype activation, Counterfactual Explanations are not only group-wise unfair with respect to race, but also group-wise unfair with respect to other demographic features including gender and age. Counterfactual Explanation has higher positive effects to induce stereotype activation for elderly female participants.

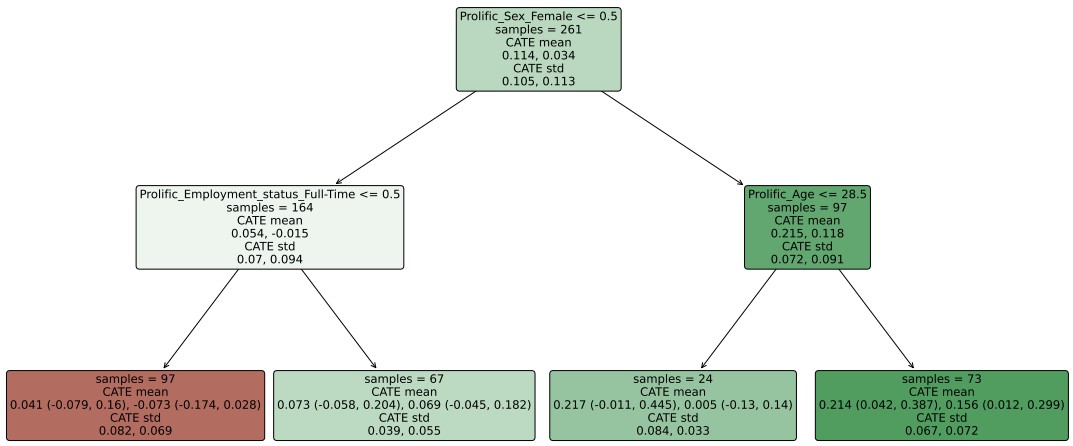

Figure 16: Counterfactual Explanation has higher positive effects to induce stereotype activation for elderly female participants.

## G  Statistically insignificant or less interesting results

We give the general results on the remaining two metrics (perceived workload, and label time) in Figure 17 and Figure 18. We give the race-specific results on the remaining metric (accuracy), where there seems to be no statistically significant evidence of disparate impact, in Figure 19. We give the general results on the number of rationales left for labeling decisions by condition and race in Figure 20 and by labeling task in Figure 21.

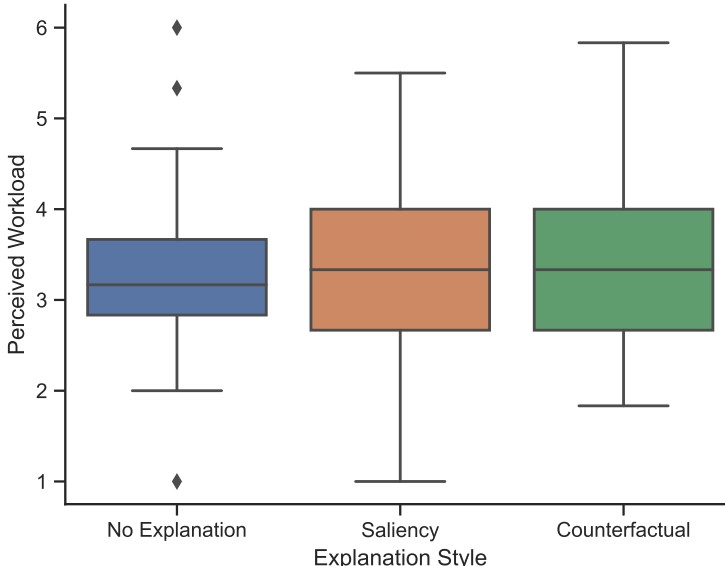

Figure 17: Perceived workload across explanation conditions

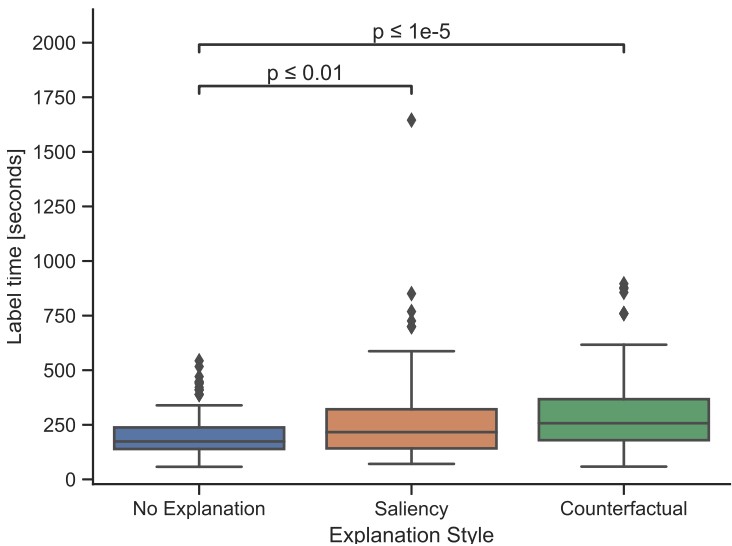

Figure 18: Label time across explanation conditions. The result that the two with-explanation conditions yield more label time than the no-explanation condition is obvious and not necessarily interesting to report as a finding.

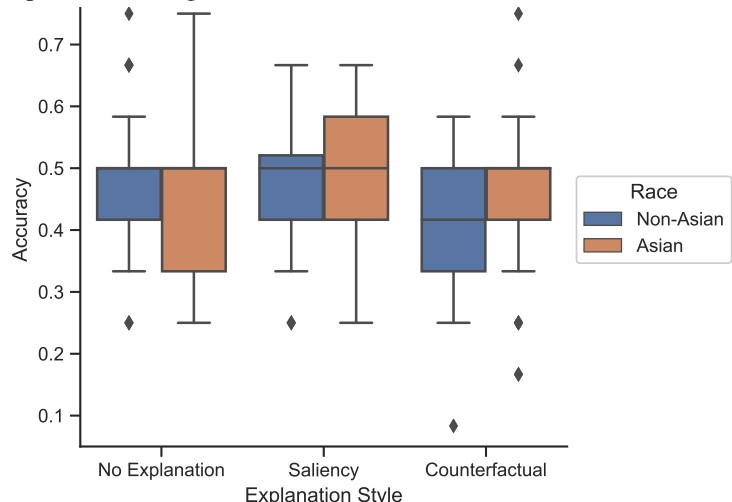

Figure 19: Accuracy across explanation conditions and racial groups.

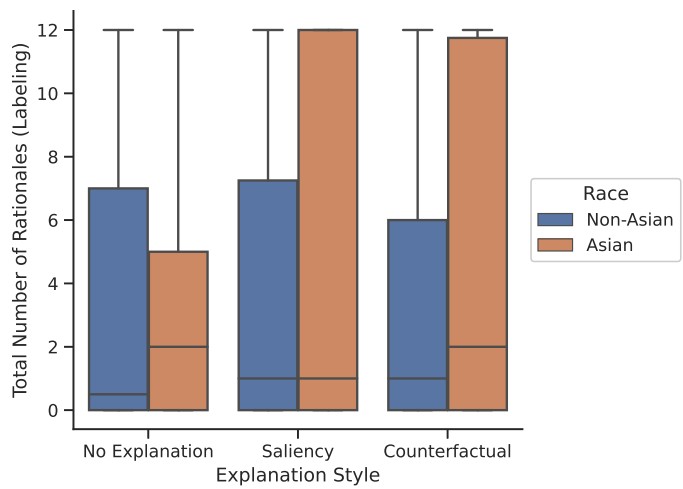

Figure 20: Number of rationales left for 12 tweet labeling questions across explanation conditions and racial groups.

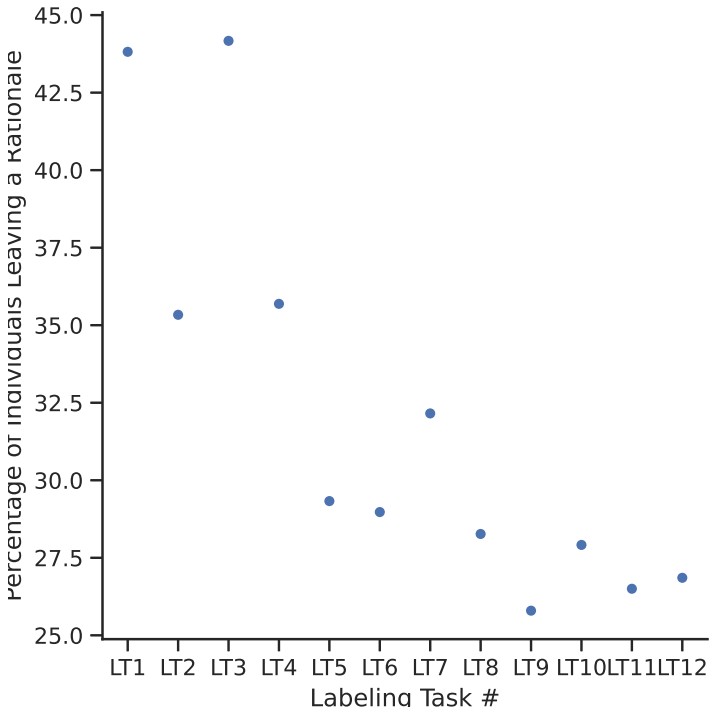

Figure 21: Percent of individuals who left a rationale for tweet labeling questions. Tweets were presented to all participants in the same order.