# OpenReview forum: "Towards Conceptualization of ``Fair Explanation'': Disparate Impacts of anti-Asian Hate Speech Explanations on Content Moderators"
_EMNLP/2023/Conference — EMNLP 2023 Main_

### Official Review · Reviewer_eKeg · 2023-08-05

**Soundness:** 2

**Excitement:**

3: Ambivalent: It has merits (e.g., it reports state-of-the-art results, the idea is nice), but there are key weaknesses (e.g., it describes incremental work), and it can significantly benefit from another round of revision. However, I won't object to accepting it if my co-reviewers champion it.

**Paper Topic And Main Contributions:**

This paper studies the psychological impact of AI generated explanations on different user groups. The paper is motivated by the need of providing fair explanation to content moderators using an AI system. The authors investigate two types of NLP explanation format: saliency maps and counterfactual explanations. To measure the impact, the authors use five metrics: classification accuracy, label time, menal discomfort, perceived workload, and stereotype activation. The results show that saliency map explanations lead to better task performance and less evidence of disparate impact than counterfactual explanations.

**Reasons To Accept:**

This paper presents results based on an online human study for fairness explanation toward specific group. This area is less investigated.
The paper proposes a new way of measuring working and mental impact for content moderators.

**Reasons To Reject:**

The paper is motivated by the need of fair AI generated explanation for content moderators in human-plus-AI system for specific group. Given such a narrowed area, I'm not sure the scope of the audience for this work.
The paper select two explanation methods without a concrete justification.




**Reproducibility:**

2: Would be hard pressed to reproduce the results. The contribution depends on data that are simply not available outside the author's institution or consortium; not enough details are provided.

**Reviewer Confidence:**

3: Pretty sure, but there's a chance I missed something. Although I have a good feel for this area in general, I did not carefully check the paper's details, e.g., the math, experimental design, or novelty.

**Typos Grammar Style And Presentation Improvements:**

Text in Figure 1 and 2 is too small and hard to read.

---

> ### Author Rebuttal · Authors · 2023-08-28
>
> Thank you so much for the time you put into reviewing our manuscript!
>
> In this **Main Rebuttal**, we will first provide a **Response** to each quoted **Concern** in your review. For **Concern 4 - Soundness**, we additionally report new results to further persuade you about the soundness of our most central finding (that counterfactual explanation is more unfair than saliency map). We only give a summary of new results in **Response 4 - Soundness**, but we provide a more detailed description of methodology and actual numbers in the **Appendix - Detailed New Results**.
>
> ## Concern 1: Scope of interest
> “Reasons To Reject: Given such a narrowed area, I'm not sure the scope of the audience for this work.”
>
> ## Response 1:
> Our paper claims two novel contributions:
>
> First, we identify an overlooked question at the intersection of the NLP explainability and fairness literature. This intersection increasingly draws more attention from the NLP community as illustrated in the literature review by Balkir et al. (2022), but all papers introduced in their review focus on how explanations improve fairness of the NLP models, and overlook the fairness of the explanations themselves. Our problem formulation opens the door to future work examining whether explanations impact people fairly in other high-stakes human-plus-AI settings inside and outside the NLP community, e.g. whether AI-generated explanations of recidivism risk assessment models yield disparate stereotype activation for bond court officials.
>
> Second, we illustrate how this “fairness of explanation” question can be answered in a **case study** (AI-assisted hate speech detection for content moderators) with statistically significant findings.
>
> Your concern about “a narrowed area” and “scope of the audience for this work” may overlook our first contribution (new research direction).
>
>
>
> ## Concern 2: Justification of explanation methods
> “Reasons To Reject: The paper selects two explanation methods without a concrete justification.”
>
> ## Response 2:
> In **Section 2. Background** of our manuscript, we did provide brief justifications for why we choose the saliency map and counterfactual explanation styles: First they contrast each other well. Second, saliency map (which tokens contribute how much to which labels) is intuitive and today’s standard saliency map method (LIME) works for any black-box classifier. Third, counterfactual explanations resemble a real-life mental process (improving hateful content).
>
> Our additional justification is that we need to choose two representative explanation methods under the big umbrella of “local explanations” (instead of “global explanation”). We need to focus on local explanation methods to explain individual predictions (tweets). We believe local explanations methods are better suited in the content moderation setting where understanding individual predictions is a priority compared to describing the average behavior of a machine learning model. Besides, local explanations methods are more helpful for content moderators to quickly digest the rationales of the model predictions and foster better human algorithm alliance.
>
>
>
> ## Concern 3: “Reproducibility: 2”
>
> ## Response 3:
> This Reproducibility score is not justified by any text in your reviews. We believe our reported results can be easily replicated since we randomly assigned participants to the three conditions, controlled the quality of responses through participant filter and attention checks, disclosed model specifications and experimental design details in the paper, and enclosed code/data files in the zip folder uploaded to OpenReview with our manuscript. Only the Personally Identifiable Information (e.g. Prolific IDs of human study participants) cannot be shared externally due to IRB regulations. We are also happy to answer any additional data or code-related questions via the “Official Comment” on OpenReview.
>
> To further assist your reviews of our code and data, below is a functionality summary of the code and data files we included in our zip folder (Supplemental Materials) as uploaded onto OpenReview.
>
> EMNLP_roberta_train.ipynb: code to fine-tune a roberta binary (hate vs. non-hate) classifier.
>
> EMNLP_roberta_test_saliency.ipynb: code to generate saliency maps (LIME) explanations.
>
> EMNLP_roberta_test_counter.ipynb: code to generate predictions for counterfactual explanations
>
> EMNLP_Calculate_all_metrics.ipynb: code to calculate 5 output metrics (accuracy, label time, mental discomfort, perceived workload, and stereotype activation).
>
> EMNLP_statistical_analysis_t_test_all_metrics_final.ipynb: code to conduct t-tests under different conditions (no explanations , saliency maps, counterfactual explanations) and to reproduce all plots included in the paper.
>
> EMNLP_Counterfactual_Explanations.xlsx: original hate-classified tweets and counterfactual explanations (human-generated counterfactual candidates classified as non-hate by the same NLP classification model)
>
> EMNLP_Human_Study_Data_with_all_Metrics.csv: processed human study data from Qualtrics and Prolific
>
>
>
> ## Concern 4: “Soundness: 2”
>
> ## Response 4:
> This Soundness score is not justified by any text in your reviews. We believe our claims are soundly supported by the empirical evidence provided in the paper because our most **central claims** (counterfactual explanations show 1. worse overall performance and 2. more unfairness evidence than saliency map) are supported by several **metric-specific claims** (mental discomfort, accuracy, stereotype activation in Section 5.1 for the overall performance claim; mental discomfort, label time, and stereotype activation in Section 5.2 for the unfairness claim). Each metric-specific claims is further supported by both **t-tests** (Figures 3-5 for overall performance; Figures 6-9 for unfairness) and **qualitative arguments from related social science papers**, e.g. we drew the connection between hetero-centric norms found in Hegarty et al. (2004) to white-dominant norms to justify unfair mental discomfort by counterfactual explanations (Section 5.2 - second paragraph).
>
> To further substantiate the soundness of our claims, we summarize **new results** from three additional streams of analysis on data from the reported human study. If you want to read the full methodology and actual numbers from each analysis stream, please check the **Appendix**.
>
> First, **Heterogeneous treatment effects** analysis with Linear Double Machine Learning (Chetverikov et al., 2016) shows that Counterfactual Explanations are not only group-wise unfair with respect to race, but also group-wise unfair with respect to other demographic features including gender and age. Second, as p-values only show statistical significance but now effect size (American Statistical Association, 2016), our **Effect size** analysis using Cohen’s d score (Cohen, 2013) shows that statistically significant effects presented in paper have small-to-medium effect sizes, necessitating statistical methods to uncover, which was what we did. Third, as we develop a simple and interpretable **Individual Fairness** pipeline based on the intuition that outcomes are individually fair if similar individuals are treated the same (Dwork et al., 2012), we find that counterfactual explanation is more individually unfair than saliency map. This new finding aligns with our previous findings in the manuscript (that counterfactual explanation shows more evidence of **group-wise unfairness**).
>
>
>
>
>
>
> ## Concern 5: Presentation improvements
> “Typos Grammar Style And Presentation Improvements: Text in Figure 1 and 2 is too small and hard to read.”
>
> ## Response 5:
> If accepted, we will increase the size of Figures 1 and 2 from double-column to single-column (larger size) in the camera-ready version. In case there is not enough space in the main text, we will provide the larger-sized figures in the camera-ready version’s appendix.
>
>
>
> Once again, we are super grateful for your time and will be even more so if you consider whether our Responses may clarify any of your Concerns!
>
>
> ### References
> 1. Esma Balkir, Svetlana Kiritchenko, Isar Nejadgholi, and Kathleen Fraser. 2022. Challenges in Applying Explainability Methods to Improve the Fairness of NLP Models. In Proceedings of the 2nd Workshop on Trustworthy Natural Language Processing (TrustNLP 2022), pages 80–92, Seattle, U.S.A.. Association for Computational Linguistics.
> 2. Chetverikov, D., Demirer, M., Duflo, E., Hansen, C., Newey, W. K., & Chernozhukov, V. (2016). Double machine learning for treatment and causal parameters. 2016.
> 3. Cohen, J. (2013). Statistical power analysis for the behavioral sciences. Academic press.
> 4. American Statistical Association. (2016). American Statistical Association releases statement on statistical significance and p-values.
> 5. Dwork, C., Hardt, M., Pitassi, T., Reingold, O., & Zemel, R. (2012, January). Fairness through awareness. In Proceedings of the 3rd innovations in theoretical computer science conference (pp. 214-226).
>
>
>
> # Appendix - Detailed New Results
>
> Please note that our decision to add new results is consistent with the EMNLP 2023 policy: “Reporting additional experiments is encouraged, this may be the most effective way to get Reviewers to increase their scores.”
>
> We report three streams of additional analysis and new results that further enhance the soundness and generalizability of our claims. Since OpenReview doesn’t allow additional uploads for author responses, we will provide the detailed code and figures associated with these new results in the camera-ready version (if accepted).
>
>
>
> ##  Heterogeneous treatment effects
> We estimate Heterogeneous treatment effects using Linear Double Machine Learning (Chetverikov et al., 2016). We use Tree Interpreter to provide a presentation-ready summary of the key features that explain the biggest differences in responsiveness to an intervention. We can provide an additional figure of the decision tree that illustrates this analysis in the camera-ready version.
>
> Specifically, we find that regarding stereotype activation, *Counterfactual Explanations are not only group-wise unfair with respect to race, but also group-wise unfair with respect to other demographic features including gender and age*. Counterfactual Explanation has higher positive effects to induce stereotype activation for elderly female participants.
>
>
>
> ## Effect size
> Since p-value can indicate whether an effect exists, it doesn't provide information about the magnitude of the effect (American Statistical Association 2016). For the readers to better interpret the substantive significance of effects that we discovered in this paper, we calculated Cohen’s d scores and 95% confidence intervals using bootstrapping.
>
> Our results indicate that *statistically significant effects presented in paper have a sensible Cohen’s d score estimates* (small to moderate effects that require statistical methods to be observed (Cohen, 2013)) and reasonable 95% confidence intervals. For example, saliency map yields significantly higher hate speech prediction accuracy than counterfactual explanation (p-value370 = 0.022, Cohen’s d: 0.337, 95% CI: [0.051, 0.625]). A Cohen’s d of 0.337 means that the two group means (Saliency maps v.s. Counterfactual Explanations) differ by 0.337 standard deviations.
>
>
>
> ## Individual Fairness
> The quantitative results we reported in the manuscript come from a group fairness perspective: whether an output feature gets significantly different values across different groups. Another popular perspective in the fairness literature is “individual fairness”, with the intuition that the outcomes are fair if similar individuals get similar outcomes (Dwork et al., 2012). If we can apply this intuition of “individual fairness” into a quantifiable metric, we might evaluate which of the three explanation conditions are more individually unfair with respect to each of the five output metrics (e.g. mental discomfort). Our motivation is that if the new “individual fairness” findings also aligns with the previous “group fairness” findings, e.g. if “counterfactual explanations” are more unfair than “saliency map” from both group fairness and individual fairness perspectives, the new result will further enhance the Soundness of our claims.
>
> One major challenge in applying the “individual fairness” intuition is to define a task-relevant pairwise distance function to calculate how “similar” any two individuals are. To address this issue, at the start of the survey, we ask each participant 9 multiple choice questions about their individual input features related to the anti-Asian hate speech classification task:
>
> Q1: Have you ever been a victim of online hate speech?
>
> …
>
> Q9: How often do you think that you "have no place in this world", "feel left out", or "feel like an outsider"?
>
> After mapping multiple-choice answers to each question to normalized values between 0 and 1, we compute the pairwise distance (absolute difference, averaged across the 9 individual input features above) for every pair of individuals within a given explanation condition. We then develop a simple and interpretable pipeline to compute an “individual unfairness” metric.
>
> Given output feature X (e.g. mental discomfort) and explanation condition i (e.g. saliency map):
>
> **Step 1**: Find the top *k* pairs of most similar two individuals (with smallest pairwise average distances)
>
> **Step 2**: Within the top k pairs, for the two individuals A and B in each pair with avg_distance(A, B), i.e. distance averaged across their 9 individual input features, to get a pairwise “individual unfairness” metric for A and B, we compute the absolute difference in their output values weighted by [(1-avg_distance(A, B)]. Our rationale is that the more similar two individuals are (i.e. the smaller their avg_distance is), the more “individually unfair” it will be for their output values to differ.
>
> **pairwise_individual_unfairness (A, B) = [1 - avg_distance(A, B)] * | output(A) - output(B) |**
>
> **Step 3**: Calculate the mean of pairwise individual unfairness scores across the k pairs to use as the individual unfairness score of explanation condition i with respect to output feature X.
>
> **Step 4**: perform t-tests between the distributions of output feature X in explanation condition i and the same output feature X in another explanation condition (such as j) to compare “individual unfairness” between conditions i and j with respect to feature X.
> We also vary the **number of top pairs considered (k)** as a hyperparameter from {100, 200, 400, 1000, 2000, 4000 pairs} and report the following findings about “individual unfairness” of the three conditions (**NE: no explanation; SM: saliency map; CE: counterfactual explanation**), which remain significant (p-value < 0.05) across multiple settings of k:
>
> 1. Stereotype Activation Individual Unfairness (SA_IU):
>
> mean SA_IU(SM) > mean SA_IU(NE) (significant when k = 1000, 2000, or 4000 pairs)
>
> mean SA_IU(CE) > mean SA_IU(SM) (significant when k = 2000, or 4000 pairs)
>
> mean SA_IU(CE) > mean SA_IU(NE) (significant when k = 1000, 2000, or 4000 pairs)
>
> Conclusion: With respect to Stereotype Activation, introducing either explanation style increases individual unfairness. Among the two styles, counterfactual explanation is more unfair.
>
> Example numbers (when k = 2000 pairs):
>
> mean SA_IU(NE) = 0.148, mean SA_IU(SM) = 0.178, mean SA_IU(CE) = 0.217
>
> p-value[SA_IU(NE), SA_IU(SM)] = 1.2 * 10^{-13}
>
> p-value[SA_IU(SM), SA_IU(CE)] = 0.00027
>
> p-value[SA_IU(CE), SA_IU(NE)] = 0.00018
>
>
>
> 2. Mental Discomfort Individual Unfairness (MD_IU):
>
> mean MD_IU(SM) < mean MD_IU(NE) (significant when k = 400, 1000, 2000, or 4000 pairs)
>
> mean MD_IU(CE) > mean MD_IU(SM) (significant when k = 400, 1000, 2000, or 4000 pairs)
>
> mean MD_IU(CE) < mean MD_IU(NE) (significant when k = 1000, 2000, or 4000 pairs)
>
> Conclusion: With respect to Mental Discomfort , introducing either explanation style decreases individual unfairness, which contrasts the individual fairness results for other output metrics.  Among the two styles, counterfactual explanation is more unfair.
>
>
>
>
>
> 3. Perceived Workload Individual Unfairness (PW_IU):
>
> mean PW_IU(SM) > mean PW_IU(NE) (significant when k = 100, 200, 400, 1000, 2000, or 4000 pairs)
>
> mean PW_IU(CE) > mean PW_IU(NE) (significant when k = 100, 200, 400, 1000, 2000, or 4000 pairs)
>
> Conclusion: With respect to Perceived Workload, introducing either explanation style increases individual unfairness.
>
>
>
>
> 4. Label time Individual Unfairness (LT_IU):
>
> mean LT_IU(SM) > mean LT_IU(NE) (significant when k = 100, 200, 400, 1000, 2000, or 4000 pairs)
>
> mean LT_IU(CE) > mean LT_IU(NE) (significant when k = 100, 200, 400, 1000, 2000, or 4000  pairs)
>
> Conclusion: With respect to Label time, introducing either explanation style increases individual unfairness. Among the two styles, counterfactual explanation is more unfair.
>
>
>
> Summary of Individual Fairness conclusions:
> - Counterfactual explanation is more individually unfair than saliency map. This new finding aligns with our previous finding in  the manuscript that counterfactual explanation shows more evidence of group-wise unfairness.
> - Introducing either explanation style increases individual unfairness with respect to many metrics (stereotype activation, perceived workload, label time) but decreases individual unfairness with respect to mental discomfort.

---

### Official Review · Reviewer_YzLM · 2023-08-05

**Soundness:** 4

**Excitement:**

4: Strong: This paper deepens the understanding of some phenomenon or lowers the barriers to an existing research direction.

**Paper Topic And Main Contributions:**

The paper studies the effect of giving a specific group (Asians) automatically generated explanations for why piece of content should have human revision (in this case Anti-Asian content). The authors compare two types of explanations (1) saliency maps generated by LIME and (2) counterfactual explanations generated by Polyjuice (based on GPT2). The authors compare the effects of these two explanations (+ no explanations) on several dimensions related to the mental states of the reviewer (time it takes to label, discomfort, etc.). They find that saliency maps outperform counterfactuals.

Main contributions in my opinion are:
- One of the first papers to investigate the effects of explainable AI on a people group
- Well-defined experiments that investigate the mental status of each group + the overall accuracy and label times
- Find that saliency maps outperform counterfactuals:
     - Asians are likelier to feel mental discomfort
     - Non-Asians likelier to take more time annotating
     - Annotators are likelier to be implicitly biased against Asians after the annotation process (Stereotype Activation)

While the paper uses relatively smaller models (RoBERTa), one of the really nice things about the paper is that it is less about the strength of the model, but rather their impact on those who see it.

**Questions For The Authors:**

Question A: 5.4 claims that Asians are less likely to leave a rationale when feeling discomfort. That is, the likelihood of an Asian-identifying annotator leaving a rationale after discomfort was 18% while the white-identifying population was 24%. What are the likelihoods of an Asian-identifying person leaving a rationale at all versus a white-identifying one? i.e, are Asians also less likely to leave a rationale all together?

Question B: How granular are the ethnicity identifiers in Prolific?

**Reasons To Accept:**

- The research question by the paper is interesting! Though I haven't quite kept up with this subfield, it is one of the first studies I have seen that compares different modes of explanability to specific groups.
- While three of the metrics are subjective (discomfort, stereotype activation, and percieved workload), I feel the authors justified them well and referred to established metrics (SPANE and NASA-TLX). In the case of stereotype activation, their formulation seemed simple and sensible.
- Experimental setup seemed clear to me and has a very clear question they want to test.

**Reasons To Reject:**

- I could see the argument that stronger models could provide better counterfactuals that may not illicit negative emotions as strongly as the ones in this paper do.
- I could see that the scope of the paper is relatively narrow.
- The above two might lead to it having less visibility/interest.

**Reproducibility:**

5: Could easily reproduce the results.

**Reviewer Confidence:**

3: Pretty sure, but there's a chance I missed something. Although I have a good feel for this area in general, I did not carefully check the paper's details, e.g., the math, experimental design, or novelty.

**Typos Grammar Style And Presentation Improvements:**

The pictures tend to be a bit small; I couldn't quite read them easily without zooming to 200%.

---

> ### Author Rebuttal · Authors · 2023-08-28
>
> Thanks for your thoughtful comments and questions! We are happy that you found our paper interesting, and our empirical findings thorough! We start our responses to address your concerns first and offer answers to your questions later.
>
> After addressing your concerns and questions, we additionally report in the **Appendix** new results to hopefully substantiate your Confidence in the strong Soundness evaluation you gave to our claims. We summarize these new results in the paragraph below for your convenience:
>
> First, **Heterogeneous treatment effects** analysis with Linear Double Machine Learning (Chetverikov et al., 2016) shows that Counterfactual Explanations are not only group-wise unfair with respect to race, but also group-wise unfair with respect to other demographic features including gender and age. Second, as p-values only show statistical significance but now effect size (American Statistical Association, 2016), our **Effect size** analysis using Cohen’s d score (Cohen, 2013) shows that statistically significant effects presented in paper have small-to-medium effect sizes, necessitating statistical methods to uncover, which was what we did. Third, as we develop a simple and interpretable **Individual Fairness** pipeline based on the intuition that outcomes are individually fair if similar individuals are treated the same (Dwork et al., 2012), we find that counterfactual explanation is more individually unfair than saliency map. This new finding aligns with our previous findings in Figure 6 and 7 of the manuscript (that counterfactual explanation shows more evidence of **group-wise unfairness**).
>
>
> ## Concern 1: Counterfactuals
> “Stronger models could provide better counterfactuals that may not illicit negative emotions as strongly as the ones in this paper do”
>
> ## Response 1
> We argue that two separate tasks should be disentangled here. Task 1: content improvement, i.e. build a model that actually improves a potentially hateful tweet into non-hateful as perceived by humans; Task 2: content (hate speech) classification. **Concern 1** is mostly related to **Task 1**. However, our focus in this paper is **Task 2** (Task 1 is good-to-have but should be a subject of an independent, algorithm-oriented project itself). Here we can simulate a very good model for Task 1 (i.e. generate completely non-hateful counterfactual candidates that differ from the original meaning of the hate-classified tweets completely), but this choice will compromise the quality of Task 2 since the counterfactual explanations will be less useful for annotators.
>
>
>
> ## Concern 2: Scope
> “The scope of the paper is relatively narrow”
>
> ## Response 2
> We agree with the argument since we mainly focus on Asian hate speech context in the case study. However, we would like to emphasize on the main contributions of this paper that it identifies an overlooked question at the intersection of the explainability and fairness literature, and illustrates that this is a question worth exploring with a case study. Our contribution opens the door to future work examining whether explanations impact people fairly in other settings.
>
>
>
> ## Concern 3: "Reproducibility: 3"
>
> “Reproducibility: 3. [...] the training/evaluation data are not widely available”
>
> ## Response 3
>
> Since the hate speech prediction model is just a tool and not a main contribution of our manuscript, we assume you find a Reproducibility weakness in our dataset. The original tweets we use come from a publicly available dataset from He et al. (2022) cited in our manuscript. You may find the link to their data in their paper’s abstract.
>
> We believe our reported results can be easily replicated since we randomly assigned participants to the three conditions, controlled the quality of responses through participant filter and attention checks, disclosed model specifications and experimental design details in the paper, and enclosed code/data files in the zip folder uploaded to OpenReview with our manuscript. Only the Personally Identifiable Information (e.g. Prolific IDs of human study participants) cannot be shared externally due to IRB regulations. We are also happy to answer any additional data or code-related questions via the “Official Comment” on OpenReview.
>
> To further assist your reviews of our code and data, below is a functionality summary of the code and data files we included in our zip folder (Supplemental Materials) as uploaded onto OpenReview.
>
> EMNLP_roberta_train.ipynb: code to fine-tune a roberta binary (hate vs. non-hate) classifier.
>
> EMNLP_roberta_test_saliency.ipynb: code to generate saliency maps (LIME) explanations.
>
> EMNLP_roberta_test_counter.ipynb: code to generate predictions for counterfactual explanations
>
> EMNLP_Calculate_all_metrics.ipynb: code to calculate 5 output metrics (accuracy, label time, mental discomfort, perceived workload, and stereotype activation).
>
> EMNLP_statistical_analysis_t_test_all_metrics_final.ipynb: code to conduct t-tests under different conditions (no explanations, saliency maps, counterfactual explanations) and to reproduce all plots included in the paper.
>
>
>
> ## Concern 4: Presentation
> “The pictures tend to be a bit small; I couldn't quite read them easily without zooming to 200%.”
>
> ## Response 4
> If accepted, we will increase the size of Figures 1 and 2 from double-column to single-column (larger size) in the camera-ready version. In case there is not enough space in the main text, we will provide the larger-sized version of all figures in the camera-ready version’s appendix.
>
>
>
>
> ## Question A:
> “What are the likelihoods of an Asian-identifying person leaving a rationale at all versus a white-identifying one? i.e, are Asians also less likely to leave a rationale all together?”
>
> ## Answer A:
> Approximately 21% of Asian-identifying people left a rationale for their mental discomfort rating versus 18% of white-identifying people. Asians are not less likely to leave a rationale altogether, but are less likely to leave one when they have experienced mental discomfort (as indicated by our quantitative metric).
>
>
>
> ## Question B:
> “How granular are the ethnicity identifiers in Prolific?”
> ## Answer B:
> As noted in our Appendix D: "we set up two almost identical Prolific studies with only one difference in the ethnicity filter: one includes only ‘Asian’ and the other includes ‘Black’, ‘White’, and ‘Other’ (i.e. non-Asian) [...] we exclude the ‘Mixed’ race category [from Prolific]." The Prolific ethnicity identifiers do not specify the participant’s ethnicity beyond Asian, White, Black, Mixed (excluded), or Other.
>
>
> Overall, we hope that these responses address any questions you had regarding the paper.
>
>
>
> ### References
> 1. Chetverikov, D., Demirer, M., Duflo, E., Hansen, C., Newey, W. K., & Chernozhukov, V. (2016). Double machine learning for treatment and causal parameters. 2016.
> 2. Cohen, J. (2013). Statistical power analysis for the behavioral sciences. Academic press.
> 3. American Statistical Association. (2016). American Statistical Association releases statement on statistical significance and p-values.
> 4. Dwork, C., Hardt, M., Pitassi, T., Reingold, O., & Zemel, R. (2012, January). Fairness through awareness. In Proceedings of the 3rd innovations in theoretical computer science conference (pp. 214-226).
> 5. Bing He, Caleb Ziems, Sandeep Soni, Naren Ramakrishnan, Diyi Yang, and Srijan Kumar. 2022. Racism is a virus: anti-asian hate and counterspeech in social media during the COVID-19 crisis. In Proceedings of the 2021 IEEE/ACM International Conference on Advances in Social Networks Analysis and Mining (ASONAM '21).
>
>
>
>
> # Appendix - Detailed New Results
>
> Please note that our decision to add new results is consistent with the EMNLP 2023 policy: “Reporting additional experiments is encouraged, this may be the most effective way to get Reviewers to increase their scores.”
>
> We report three streams of additional analysis and new results that further enhance the soundness and generalizability of our claims. Since OpenReview doesn’t allow additional uploads for author responses, we will provide the detailed code and figures associated with these new results in the camera-ready version (if accepted).
>
>
>
> ##  Heterogeneous treatment effects
> We estimate Heterogeneous treatment effects using Linear Double Machine Learning (Chetverikov et al., 2016). We use Tree Interpreter to provide a presentation-ready summary of the key features that explain the biggest differences in responsiveness to an intervention. We can provide an additional figure of the decision tree that illustrates this analysis in the camera-ready version.
>
> Specifically, we find that regarding stereotype activation, *Counterfactual Explanations are not only group-wise unfair with respect to race, but also group-wise unfair with respect to other demographic features including gender and age*. Counterfactual Explanation has higher positive effects to induce stereotype activation for elderly female participants.
>
>
>
> ## Effect size
> Since p-value can indicate whether an effect exists, it doesn't provide information about the magnitude of the effect (American Statistical Association 2016). For the readers to better interpret the substantive significance of effects that we discovered in this paper, we calculated Cohen’s d scores and 95% confidence intervals using bootstrapping.
>
> Our results indicate that *statistically significant effects presented in paper have a sensible Cohen’s d score estimates* (small to moderate effects that require statistical methods to be observed (Cohen, 2013)) and reasonable 95% confidence intervals. For example, saliency map yields significantly higher hate speech prediction accuracy than counterfactual explanation (p-value370 = 0.022, Cohen’s d: 0.337, 95% CI: [0.051, 0.625]). A Cohen’s d of 0.337 means that the two group means (Saliency maps v.s. Counterfactual Explanations) differ by 0.337 standard deviations.
>
>
>
> ## Individual Fairness
> The quantitative results we reported in the manuscript come from a group fairness perspective: whether an output feature gets significantly different values across different groups. Another popular perspective in the fairness literature is “individual fairness”, with the intuition that the outcomes are fair if similar individuals get similar outcomes (Dwork et al., 2012). If we can apply this intuition of “individual fairness” into a quantifiable metric, we might evaluate which of the three explanation conditions are more individually unfair with respect to each of the five output metrics (e.g. mental discomfort). Our motivation is that if the new “individual fairness” findings also aligns with the previous “group fairness” findings, e.g. if “counterfactual explanations” are more unfair than “saliency map” from both group fairness and individual fairness perspectives, the new result will further enhance the Soundness of our claims.
>
> One major challenge in applying the “individual fairness” intuition is to define a task-relevant pairwise distance function to calculate how “similar” any two individuals are. To address this issue, at the start of the survey, we ask each participant 9 multiple choice questions about their individual input features related to the anti-Asian hate speech classification task:
>
> Q1: Have you ever been a victim of online hate speech?
>
> …
>
> Q9: How often do you think that you "have no place in this world", "feel left out", or "feel like an outsider"?
>
> After mapping multiple-choice answers to each question to normalized values between 0 and 1, we compute the pairwise distance (absolute difference, averaged across the 9 individual input features above) for every pair of individuals within a given explanation condition. We then develop a simple and interpretable pipeline to compute an “individual unfairness” metric.
>
> Given output feature X (e.g. mental discomfort) and explanation condition i (e.g. saliency map):
>
> **Step 1**: Find the top *k* pairs of most similar two individuals (with smallest pairwise average distances)
>
> **Step 2**: Within the top k pairs, for the two individuals A and B in each pair with avg_distance(A, B), i.e. distance averaged across their 9 individual input features, to get a pairwise “individual unfairness” metric for A and B, we compute the absolute difference in their output values weighted by [(1-avg_distance(A, B)]. Our rationale is that the more similar two individuals are (i.e. the smaller their avg_distance is), the more “individually unfair” it will be for their output values to differ.
>
> **pairwise_individual_unfairness (A, B) = [1 - avg_distance(A, B)] * | output(A) - output(B) |**
>
> **Step 3**: Calculate the mean of pairwise individual unfairness scores across the k pairs to use as the individual unfairness score of explanation condition i with respect to output feature X.
>
> **Step 4**: perform t-tests between the distributions of output feature X in explanation condition i and the same output feature X in another explanation condition (such as j) to compare “individual unfairness” between conditions i and j with respect to feature X.
> We also vary the **number of top pairs considered (k)** as a hyperparameter from {100, 200, 400, 1000, 2000, 4000 pairs} and report the following findings about “individual unfairness” of the three conditions (**NE: no explanation; SM: saliency map; CE: counterfactual explanation**), which remain significant (p-value < 0.05) across multiple settings of k:
>
> 1. Stereotype Activation Individual Unfairness (SA_IU):
>
> mean SA_IU(SM) > mean SA_IU(NE) (significant when k = 1000, 2000, or 4000 pairs)
>
> mean SA_IU(CE) > mean SA_IU(SM) (significant when k = 2000, or 4000 pairs)
>
> mean SA_IU(CE) > mean SA_IU(NE) (significant when k = 1000, 2000, or 4000 pairs)
>
> Conclusion: With respect to Stereotype Activation, introducing either explanation style increases individual unfairness. Among the two styles, counterfactual explanation is more unfair.
>
> Example numbers (when k = 2000 pairs):
>
> mean SA_IU(NE) = 0.148, mean SA_IU(SM) = 0.178, mean SA_IU(CE) = 0.217
>
> p-value[SA_IU(NE), SA_IU(SM)] = 1.2 * 10^{-13}
>
> p-value[SA_IU(SM), SA_IU(CE)] = 0.00027
>
> p-value[SA_IU(CE), SA_IU(NE)] = 0.00018
>
>
>
> 2. Mental Discomfort Individual Unfairness (MD_IU):
>
> mean MD_IU(SM) < mean MD_IU(NE) (significant when k = 400, 1000, 2000, or 4000 pairs)
>
> mean MD_IU(CE) > mean MD_IU(SM) (significant when k = 400, 1000, 2000, or 4000 pairs)
>
> mean MD_IU(CE) < mean MD_IU(NE) (significant when k = 1000, 2000, or 4000 pairs)
>
> Conclusion: With respect to Mental Discomfort , introducing either explanation style decreases individual unfairness, which contrasts the individual fairness results for other output metrics.  Among the two styles, counterfactual explanation is more unfair.
>
>
>
>
>
> 3. Perceived Workload Individual Unfairness (PW_IU):
>
> mean PW_IU(SM) > mean PW_IU(NE) (significant when k = 100, 200, 400, 1000, 2000, or 4000 pairs)
>
> mean PW_IU(CE) > mean PW_IU(NE) (significant when k = 100, 200, 400, 1000, 2000, or 4000 pairs)
>
> Conclusion: With respect to Perceived Workload, introducing either explanation style increases individual unfairness.
>
>
>
>
> 4. Label time Individual Unfairness (LT_IU):
>
> mean LT_IU(SM) > mean LT_IU(NE) (significant when k = 100, 200, 400, 1000, 2000, or 4000 pairs)
>
> mean LT_IU(CE) > mean LT_IU(NE) (significant when k = 100, 200, 400, 1000, 2000, or 4000  pairs)
>
> Conclusion: With respect to Label time, introducing either explanation style increases individual unfairness. Among the two styles, counterfactual explanation is more unfair.
>
>
>
> Summary of Individual Fairness conclusions:
> - Counterfactual explanation is more individually unfair than saliency map. This new finding aligns with our previous finding in  the manuscript that counterfactual explanation shows more evidence of group-wise unfairness.
> - Introducing either explanation style increases individual unfairness with respect to many metrics (stereotype activation, perceived workload, label time) but decreases individual unfairness with respect to mental discomfort.

---

### Official Review · Reviewer_JNom · 2023-08-05

**Soundness:** 3

**Excitement:**

4: Strong: This paper deepens the understanding of some phenomenon or lowers the barriers to an existing research direction.

**Paper Topic And Main Contributions:**

The authors investigate the disparate impact of two forms of explanations, namely saliency maps and counterfactuals, in the context of hate speech detection on a dataset targeting anti-Asian hate. They analyze 12 tweets, manually crafting the counterfactual explanations, and employing LIME to generate saliency maps. The study also assesses a pool of factors, including mental discomfort, stereotype activation, accuracy, and perceived workload.

**Reasons To Accept:**

This paper studies a novel impact of explanations on annotators and specifically takes into account the details about annotators such as their identity that have been shown previously to impact their annotation behavior on the task of hate speech detection.


**Reasons To Reject:**

My primary concern with this study is that the small sample size of 12 tweets may not provide a robust foundation for comparing the two explanation methods, saliency maps and counterfactuals. If the goal is to understand the disparate impact of explanations on moderators, it would be beneficial to expand the scope of the investigation to include other forms of explanations, such as gradient-based, attention-based, or global ones. Furthermore, the paper's group-specific conclusions are confined solely to anti-Asian hate, limiting its broader applicability. I encourage the authors to replicate these findings with other social groups, such as the African-American community, to enhance the generalizability and depth of their insights.

**Reproducibility:**

4: Could mostly reproduce the results, but there may be some variation because of sample variance or minor variations in their interpretation of the protocol or method.

**Reviewer Confidence:**

4: Quite sure. I tried to check the important points carefully. It's unlikely, though conceivable, that I missed something that should affect my ratings.

---

> ### Author Rebuttal · Authors · 2023-08-28
>
> Thank you so much for the time you put into reviewing our manuscript!
>
> In this **Main Rebuttal**, we will first provide a **Response** to each quoted **Concern** in your review. For **Concern 4 - Soundness**, we additionally report new results to further persuade you about the soundness of our most central finding (that counterfactual explanation is more unfair than saliency map). We only give a summary of new results in **Response 4 - Soundness**, but we provide a more detailed description of methodology and actual numbers in the **Appendix - Detailed New Results**.
>
>
>
> ## Concern 1: Small sample size
> “My primary concern with this study is that the small sample size of 12 tweets may not provide a robust foundation for comparing the two explanation methods, saliency maps and counterfactuals.”
>
> ## Response 1:
> We first argue the rationales to choose a relatively small sample size of tweets. First, we choose the sample size such that it is more likely for participants to better focus on the assigned task, circumventing potential information overload problems. It has been acknowledged that information overload can deteriorate decision making quality, prolong the decision process, and make participants less confident about their choices (Chervany, N. L., & Dickson, G. W. ,1974; O'Reilly III, C. A.,1980; Hwang, M. I., & Lin, J. W.,1999). Furthermore, we emphasize on the trade-off between the number of participants vs. task length under a fixed budget, we must not compromise the former if we want a reasonable sample size of participants.
>
> We agree that it is better to determine a reasonable number of tweets that can both offer a robust foundation for comparing explanation methods and won’t introduce information overload problems that would compromise quality of the responses from participants.
>
> ## Concern 2:  Other forms of explanations
> “If the goal is to understand the disparate impact of explanations on moderators, it would be beneficial to expand the scope of the investigation to include other forms of explanations, such as gradient-based, attention-based, or global ones”
>
> ## Response 2:
> We appreciate your suggestions to include other explanation methods to expand the scope of research. It is definitely worthwhile exploring for future studies. However, we would like to caution the readers that some explanation methods mentioned in the comments are not very appropriate to this hate speech moderation context.
>
> First, the global explanations aim to describe the average behavior of a machine learning model rather than explain individual predictions. We believe local explanations methods are better suited in the content moderation setting where understanding individual predictions is a priority.
> Furthermore, gradient-based methods or attention methods require solid background knowledge to understand which may impose challenges for participants to interpret and thus make inferior decisions with poor understanding.
>
> In essence, we choose the two explanation methods because LIME-based saliency map is straightforward, intuitive, and works for any black-box classifier while counterfactual explanation resembles a real-life mental process. It is possible to include other appropriate explanation methods to strengthen the findings in this paper or offer more insights but we think it is not the main focus of this paper and we leave it for future research.
>
>
>
> ## Concern 3:  Generalizability and depth of insights
> “Furthermore, the paper's group-specific conclusions are confined solely to anti-Asian hate, limiting its broader applicability. I encourage the authors to replicate these findings with other social groups, such as the African-American community, to enhance the generalizability and depth of their insights.”
>
> ## Response 3:
> We definitely agree with your suggestions. We think that future work may investigate the fairness of explanation in the same context but with respect to a different group (e.g. hate speech against African-Americans), a different feature (e.g. gender instead of race), or a different human-plus-AI context (e.g. disparate impact on bond court official from different groups by feature importance weights as explanations for recidivism risk assessment models, which may bias against people of colors).
>
> Within the limit of our study’s budget, we tried our best to enhance the generalizability of our findings by conducting further statistical analysis. Specifically, we calculate Cohen’s scores to measure the magnitude of the effects reported in this paper and use Linear Double Machine Learning model to estimate the Heterogeneous treatment effects to offer more sights and justification about our findings. Specifically, we find that regarding stereotype activation, Counterfactual Explanations are not only group-wise unfair with respect to race, but also group-wise unfair with respect to other demographic features including gender and age. Counterfactual Explanation has higher positive effects to induce stereotype activation for elderly female participants.
>
>
>
> ## Concern 4: “Soundness: 3”
>
> ## Response 4:
> As your Soundness score suggests that “some minor points may need extra support or details”, to further substantiate the soundness of our claims, we summarize **new results** from three additional streams of analysis on data from the reported human study. If you want to read the full methodology and actual numbers from each analysis stream, please check the **Appendix** at the end of this our author response.
>
> First, **Heterogeneous treatment effects** analysis with Linear Double Machine Learning (Chetverikov et al., 2016) shows that Counterfactual Explanations are not only group-wise unfair with respect to race, but also group-wise unfair with respect to other demographic features including gender and age. Second, as p-values only show statistical significance but now effect size (American Statistical Association, 2016), our **Effect size** analysis using Cohen’s d score (Cohen, 2013) shows that statistically significant effects presented in paper have small-to-medium effect sizes, necessitating statistical methods to uncover, which was what we did. Third, as we develop a simple and interpretable **Individual Fairness** pipeline based on the intuition that outcomes are individually fair if similar individuals are treated the same (Dwork et al., 2012), we find that counterfactual explanation is more individually unfair than saliency map. This new finding aligns with our previous findings in Figure 6 and 7 of the manuscript (that counterfactual explanation shows more evidence of **group-wise unfairness**).
>
>
>
> Overall, we hope that these responses address any questions you had regarding the paper.
>
> ### References
> 1. Chervany, N. L., & Dickson, G. W. (1974). An experimental evaluation of information overload in a production environment. Management science, 20(10), 1335-1344.
> 2. O'Reilly III, C. A. (1980). Individuals and information overload in organizations: is more necessarily better?. Academy of management journal, 23(4), 684-696.
> 3. Hwang, M. I., & Lin, J. W. (1999). Information dimension, information overload and decision quality. Journal of information science, 25(3), 213-218.
> 4. Chetverikov, D., Demirer, M., Duflo, E., Hansen, C., Newey, W. K., & Chernozhukov, V. (2016). Double machine learning for treatment and causal parameters. 2016.
> 5. Cohen, J. (2013). Statistical power analysis for the behavioral sciences. Academic press.
> 6. American Statistical Association. (2016). American Statistical Association releases statement on statistical significance and p-values.
> 7. Dwork, C., Hardt, M., Pitassi, T., Reingold, O., & Zemel, R. (2012, January). Fairness through awareness. In Proceedings of the 3rd innovations in theoretical computer science conference (pp. 214-226).
>
>
>
> # Appendix - Detailed New Results
>
> Please note that our decision to add new results is consistent with the EMNLP 2023 policy: “Reporting additional experiments is encouraged, this may be the most effective way to get Reviewers to increase their scores.”
>
> We report three streams of additional analysis and new results that further enhance the soundness and generalizability of our claims. Since OpenReview doesn’t allow additional uploads for author responses, we will provide the detailed code and figures associated with these new results in the camera-ready version (if accepted).
>
>
>
> ##  Heterogeneous treatment effects
> We estimate Heterogeneous treatment effects using Linear Double Machine Learning (Chetverikov et al., 2016). We use Tree Interpreter to provide a presentation-ready summary of the key features that explain the biggest differences in responsiveness to an intervention. We can provide an additional figure of the decision tree that illustrates this analysis in the camera-ready version.
>
> Specifically, we find that regarding stereotype activation, *Counterfactual Explanations are not only group-wise unfair with respect to race, but also group-wise unfair with respect to other demographic features including gender and age*. Counterfactual Explanation has higher positive effects to induce stereotype activation for elderly female participants.
>
>
>
> ## Effect size
> Since p-value can indicate whether an effect exists, it doesn't provide information about the magnitude of the effect (American Statistical Association 2016). For the readers to better interpret the substantive significance of effects that we discovered in this paper, we calculated Cohen’s d scores and 95% confidence intervals using bootstrapping.
>
> Our results indicate that *statistically significant effects presented in paper have a sensible Cohen’s d score estimates* (small to moderate effects that require statistical methods to be observed (Cohen, 2013)) and reasonable 95% confidence intervals. For example, saliency map yields significantly higher hate speech prediction accuracy than counterfactual explanation (p-value370 = 0.022, Cohen’s d: 0.337, 95% CI: [0.051, 0.625]). A Cohen’s d of 0.337 means that the two group means (Saliency maps v.s. Counterfactual Explanations) differ by 0.337 standard deviations.
>
>
>
> ## Individual Fairness
> The quantitative results we reported in the manuscript come from a group fairness perspective: whether an output feature gets significantly different values across different groups. Another popular perspective in the fairness literature is “individual fairness”, with the intuition that the outcomes are fair if similar individuals get similar outcomes (Dwork et al., 2012). If we can apply this intuition of “individual fairness” into a quantifiable metric, we might evaluate which of the three explanation conditions are more individually unfair with respect to each of the five output metrics (e.g. mental discomfort). Our motivation is that if the new “individual fairness” findings also aligns with the previous “group fairness” findings, e.g. if “counterfactual explanations” are more unfair than “saliency map” from both group fairness and individual fairness perspectives, the new result will further enhance the Soundness of our claims.
>
> One major challenge in applying the “individual fairness” intuition is to define a task-relevant pairwise distance function to calculate how “similar” any two individuals are. To address this issue, at the start of the survey, we ask each participant 9 multiple choice questions about their individual input features related to the anti-Asian hate speech classification task:
>
> Q1: Have you ever been a victim of online hate speech?
>
> …
>
> Q9: How often do you think that you "have no place in this world", "feel left out", or "feel like an outsider"?
>
> After mapping multiple-choice answers to each question to normalized values between 0 and 1, we compute the pairwise distance (absolute difference, averaged across the 9 individual input features above) for every pair of individuals within a given explanation condition. We then develop a simple and interpretable pipeline to compute an “individual unfairness” metric.
>
> Given output feature X (e.g. mental discomfort) and explanation condition i (e.g. saliency map):
>
> **Step 1**: Find the top *k* pairs of most similar two individuals (with smallest pairwise average distances)
>
> **Step 2**: Within the top k pairs, for the two individuals A and B in each pair with avg_distance(A, B), i.e. distance averaged across their 9 individual input features, to get a pairwise “individual unfairness” metric for A and B, we compute the absolute difference in their output values weighted by [(1-avg_distance(A, B)]. Our rationale is that the more similar two individuals are (i.e. the smaller their avg_distance is), the more “individually unfair” it will be for their output values to differ.
>
> **pairwise_individual_unfairness (A, B) = [1 - avg_distance(A, B)] * | output(A) - output(B) |**
>
> **Step 3**: Calculate the mean of pairwise individual unfairness scores across the k pairs to use as the individual unfairness score of explanation condition i with respect to output feature X.
>
> **Step 4**: perform t-tests between the distributions of output feature X in explanation condition i and the same output feature X in another explanation condition (such as j) to compare “individual unfairness” between conditions i and j with respect to feature X.
> We also vary the **number of top pairs considered (k)** as a hyperparameter from {100, 200, 400, 1000, 2000, 4000 pairs} and report the following findings about “individual unfairness” of the three conditions (**NE: no explanation; SM: saliency map; CE: counterfactual explanation**), which remain significant (p-value < 0.05) across multiple settings of k:
>
> 1. Stereotype Activation Individual Unfairness (SA_IU):
>
> mean SA_IU(SM) > mean SA_IU(NE) (significant when k = 1000, 2000, or 4000 pairs)
>
> mean SA_IU(CE) > mean SA_IU(SM) (significant when k = 2000, or 4000 pairs)
>
> mean SA_IU(CE) > mean SA_IU(NE) (significant when k = 1000, 2000, or 4000 pairs)
>
> Conclusion: With respect to Stereotype Activation, introducing either explanation style increases individual unfairness. Among the two styles, counterfactual explanation is more unfair.
>
> Example numbers (when k = 2000 pairs):
>
> mean SA_IU(NE) = 0.148, mean SA_IU(SM) = 0.178, mean SA_IU(CE) = 0.217
>
> p-value[SA_IU(NE), SA_IU(SM)] = 1.2 * 10^{-13}
>
> p-value[SA_IU(SM), SA_IU(CE)] = 0.00027
>
> p-value[SA_IU(CE), SA_IU(NE)] = 0.00018
>
>
>
> 2. Mental Discomfort Individual Unfairness (MD_IU):
>
> mean MD_IU(SM) < mean MD_IU(NE) (significant when k = 400, 1000, 2000, or 4000 pairs)
>
> mean MD_IU(CE) > mean MD_IU(SM) (significant when k = 400, 1000, 2000, or 4000 pairs)
>
> mean MD_IU(CE) < mean MD_IU(NE) (significant when k = 1000, 2000, or 4000 pairs)
>
> Conclusion: With respect to Mental Discomfort , introducing either explanation style decreases individual unfairness, which contrasts the individual fairness results for other output metrics.  Among the two styles, counterfactual explanation is more unfair.
>
>
>
>
>
> 3. Perceived Workload Individual Unfairness (PW_IU):
>
> mean PW_IU(SM) > mean PW_IU(NE) (significant when k = 100, 200, 400, 1000, 2000, or 4000 pairs)
>
> mean PW_IU(CE) > mean PW_IU(NE) (significant when k = 100, 200, 400, 1000, 2000, or 4000 pairs)
>
> Conclusion: With respect to Perceived Workload, introducing either explanation style increases individual unfairness.
>
>
>
>
> 4. Label time Individual Unfairness (LT_IU):
>
> mean LT_IU(SM) > mean LT_IU(NE) (significant when k = 100, 200, 400, 1000, 2000, or 4000 pairs)
>
> mean LT_IU(CE) > mean LT_IU(NE) (significant when k = 100, 200, 400, 1000, 2000, or 4000  pairs)
>
> Conclusion: With respect to Label time, introducing either explanation style increases individual unfairness. Among the two styles, counterfactual explanation is more unfair.
>
>
>
> Summary of Individual Fairness conclusions:
> - Counterfactual explanation is more individually unfair than saliency map. This new finding aligns with our previous finding in  the manuscript that counterfactual explanation shows more evidence of group-wise unfairness.
> - Introducing either explanation style increases individual unfairness with respect to many metrics (stereotype activation, perceived workload, label time) but decreases individual unfairness with respect to mental discomfort.

---

### Meta-Review · Area_Chair_2Kaj · 2023-09-15

**Recommendation:** 5

**Metareview:**

The reviewers found the paper interesting, exciting, and sound. The authors addressed the issues that were brought up in the discussion phase. The only unaddressed criticism is related to the potentially narrow scope.

---

### Decision · Program_Chairs · 2023-10-07

**Decision:**

Accept-Main

**Comment:**

The reviewers found the paper interesting, exciting, and sound. The authors addressed the issues that were brought up in the discussion phase. The only unaddressed criticism is related to the potentially narrow scope.